# Renal UTX-PHGDH-serine axis regulates metabolic disorders in the kidney and liver

Hong Chen 🄳 [1], Chong Liu[2], Qian Wang[1], Mingrui Xiong[1], Xia Zeng[1], Dong Yang[1], Yunhao Xie[2], Hua Su[3], Yu Zhang[1], Yixue Huang[1], Yuchen Chen[1], Junqiu Yue[4], Chengyu Liu[5], Shun Wang[5], Kun Huang 🄳 [1✉] & Ling Zheng 🄳 [2,6✉]

Global obesity epidemics impacts human health and causes obesity-related illnesses, including the obesity-related kidney and liver diseases. UTX, a histone H3K27 demethylase, plays important roles in development and differentiation. Here we show that kidney-specific knockout *Utx* inhibits high-fat diet induced lipid accumulation in the kidney and liver via upregulating circulating serine levels. Mechanistically, UTX recruits E3 ligase RNF114 to ubiquitinate phosphoglycerate dehydrogenase, the rate limiting enzyme for de novo serine synthesis, at Lys[310] and Lys[330], which leads to its degradation, and thus suppresses renal and circulating serine levels. Consistently, phosphoglycerate dehydrogenase and serine levels are markedly downregulated in human subjects with diabetic kidney disease or obesity-related renal dysfunction. Notably, oral administration of serine ameliorates high-fat diet induced fatty liver and renal dysfunction, suggesting a potential approach against obesity related metabolic disorders. Together, our results reveal a metabolic homeostasis regulation mediated by a renal UTX-PHGDH-serine axis.

[1] Tongji School of Pharmacy, Tongji Medical College, Huazhong University of Science and Technology, 430030 Wuhan, China. [2] Hubei Key Laboratory of Cell Homeostasis, College of Life Sciences, Wuhan University, 430072 Wuhan, China. [3] Department of Nephrology, Union Hospital, Tongji Medical College, Huazhong University of Science and Technology, 430030 Wuhan, China. [4] Department of Pathology, Hubei Cancer Hospital, Tongji Medical College, Huazhong University of Science and Technology, 430030 Wuhan, China. [5] Department of Blood Transfusion, Wuhan Hospital of Traditional and Western Medicine, Wuhan 430022, China. [6] Frontier Science Center for Immunology and Metabolism, Wuhan University, 430072 Wuhan, China. ✉email: kunhuang@hust.edu.cn; lzheng@whu.edu.cn

Obesity, which results from an imbalance between caloric intake and energy expenditure, is a severe public health problem, affecting near 2 billion population worldwide[1–3]. Obesity is associated with increased risk of chronic kidney diseases, which eventually progress to end-stage renal disease (ESRD)[4,5]. Compared with normal weight subjects, the hazard ratios of ESRD risk are 1.87 and 3.57 for those with overweight or obesity[5,6].

Abnormalities, including glomerulomegaly, mesangial expansion, tubular atrophy, and renal interstitial fibrosis, have been observed in individuals with obesity[4]. Moreover, ectopic lipid accumulation is found in the kidney, especially in the tubular cells, of human and mouse with obesity[4]. Therefore, similar to non-alcoholic fatty liver disease (NAFLD), obesity-related kidney disease (also known as fatty kidney disease), has been recognized as a complication for obesity[4]. Lipid accumulation in the kidney occurs in both metabolic and non-metabolic origin, as found in diabetic kidney disease (DKD), focal segmental glomerulo-sclerosis, adriamycin-induced nephropathy and Alport syndrome[7–10]. Moreover, increased absorption of albumin-bound fatty acids by proximal tubular cells leads to tubular insulin resistance and enhanced renal gluconeogenesis, eventually causes tubular atrophy[4,11]. However, how fatty kidney develops remains not fully understood.

The most well-known function of kidney is to maintain acid-base balance and electrolyte homeostasis, and to remove waste products; however, it also plays important endocrine roles by releasing a number of important factors, such as renin, ery-thropoietin, prostaglandin and aldosterone, thus involved in regulating bone growth, cardiovascular system, and hematopoiesis[12,13]. A crosstalk between kidney and white adipose tissue has been revealed in a mouse model of chronic kidney disease (CKD), where renal lesion increases the circulating levels of parathyroid hormone and parathyroid hormone-related pro-tein, which promotes their binding to the receptor PTHR in the white adipose tissue, and stimulates browning[14]. However, con-nections between kidney and other metabolic organs remain not fully understood.

UTX (ubiquitously transcribed tetratricopeptide repeat on chromosome X, also known as KDM6A), a H3K27 demethylase, plays roles in various biological processes, such as controlling cell fate, embryonic development, and cancers[15]. Recently, upregu-lation of UTX has been found in the kidneys of patients with diabetic kidney disease or focal segmental glomerulosclerosis, as well as in the kidneys of mouse models of type 1 diabetes (Akita and OVE mouse) and type 2 diabetes (db/db mouse)[16–18], indi-cating an important role of UTX in the development of diabetic kidney disease. However, the roles of renal UTX in obesity-related diseases remain unclear.

Here, we report that kidney-specific knockout Utx reduces renal and hepatic steatosis under HFD stress. Notably, renal UTX recruits E3 ligase RNF114, which leads to PHGDH degradation and suppression of serine synthesis. Consistently, oral serine treatment ameliorated HFD-induced obesity-related kidney dis-ease and NAFLD. Together, our results reveal that a renal UTX-PHGDH-serine axis regulates metabolic homeostasis in the kid-ney and liver.

## Results

**UTX is upregulated in obesity-related kidney disease.** Previous studies by others and us have reported increased renal UTX expression in patients with diabetic kidney disease or focal seg-mental glomerulosclerosis, as well as in the kidneys of diabetic mouse models including Akita mice, db/db mice, OVE26 mice, and STZ-treated mice[16–18]. Herein, we investigated the role of

UTX in obesity-related kidney disease. In human tissues, com-pared to the controls, patients with obesity showed dramatically upregulated UTX in renal tubular and glomerular cells (Fig. 1a and Supplementary Table 1). Consistently, upregulated UTX was observed in the kidney of HFD-fed mouse, further co-immunofluorescence staining demonstrated elevated UTX in the nuclei of proximal tubular cells, distal tubules, mesangial cells and podocytes (Fig. 1b).

**Kidney-specific *Utx* knockout mice show a lean phenotype under HFD stress.** To study the role of UTX in the kidney, we generated two strains of kidney-specific *Utx* knockout mice. Significantly reduced mRNA and protein levels of UTX, as well as increased H3K27me3 level in whole kidney or renal medulla and cortex, but not in the liver and brown adipose tissue (BAT) (Fig. 1c–e, Supplementary Fig. 1a–f), were found in both $Utx^{Pax2}$ KO and $Utx^{Ksp}$ KO mice, indicating kidney-specific knockout of *Utx*. The *Utx* KO mice were generally healthy and normally born, with no obvious behavioral abnormality.

To study the role of renal UTX in obesity related kidney diseases, $Utx^{Pax2}$ KO and $Utx^{Ksp}$ KO mice, as well as their respective WT littermates, were challenged with HFD as indicated (Supplementary Fig. 1g). Under HFD stress, decreased UTX level was observed in the tubular cells of $Utx^{Ksp}$ KO mice (Fig. 1f). Surprisingly, under HFD feeding, compared with WT mice, both $Utx^{Ksp}$ KO and $Utx^{Pax2}$ KO mice showed significantly lower body weights, reduced fat, increased lean mass, as well as reduced weights of liver, inguinal white adipose tissue (iWAT) and brown adipose tissue (BAT) (Fig. 1g–i, Supplementary Figs. 1h and 2a–c, Supplementary Tables 2 and 3), even though $Utx^{Ksp}$ KO mice showed slightly higher amount of food intake at some time points (Supplementary Fig. 1i). Compared to HFD-fed WT mice, HFD-fed $Utx^{Pax2}$ KO mice also showed significantly lower kidney weights (Supplementary Table 3).

Furthermore, consistent with their lean phenotypes under the HFD stress, $Utx^{Ksp}$ KO or $Utx^{Pax2}$ KO mice showed significantly decreased serum triglyceride (TG), cholesterol, leptin, insulin, and non-fasting blood glucose (NFBG) levels (Fig. 1j–l and Supple-mentary Fig. 2d–g), as well as significantly improved glucose tolerance and insulin tolerance (Fig. 1m, n and Supplementary Figs. 1j, k and 2h, i). Meanwhile, the levels of uBUN (urine blood urea nitrogen) and uACR (urine albumin/creatinine ratio), which are commonly used to monitor the development and extent of renal damage[19–21], were significantly elevated in HFD-fed WT mice. $Utx^{Ksp}$ KO showed a trend of inhibition on uACR level under HFD-fed conditions, with similar eGFR (estimated glomerular filtration rate) level in these groups (Supplementary Fig. 1l–n). Interestingly, liver or adipose tissue specific knockout *Utx* in male mice showed no obvious effect on HFD-induced obesity (Supplementary Fig. 2j, k).

**Kidney-specific *Utx* knockout ameliorates obesity-induced renal morphological disorders.** In a murine obesity-related kidney disease model, HFD feeding induced glomerular expan-sion, mesangial matrix expansion and lipid accumulation in the kidney[22,23]. Compared with respective HFD-fed WT mice, both HFD-fed $Utx^{Ksp}$ KO and $Utx^{Pax2}$ KO mice showed significantly decreased glomerular hypertrophy and lipid droplet formation in renal tubules (demonstrated by H&E staining), while decreased mesangial expansion was detected in HFD-fed $Utx^{Ksp}$ KO mice (demonstrated by PASH staining), indicating less morphological injury in the kidney under HFD stress (Fig. 2a, b and Supple-mentary Fig. 3a, b). Furthermore, significantly decreased levels of TG and cholesterol were found in both medulla and cortex, as well as significantly downregulated transcription levels of genes

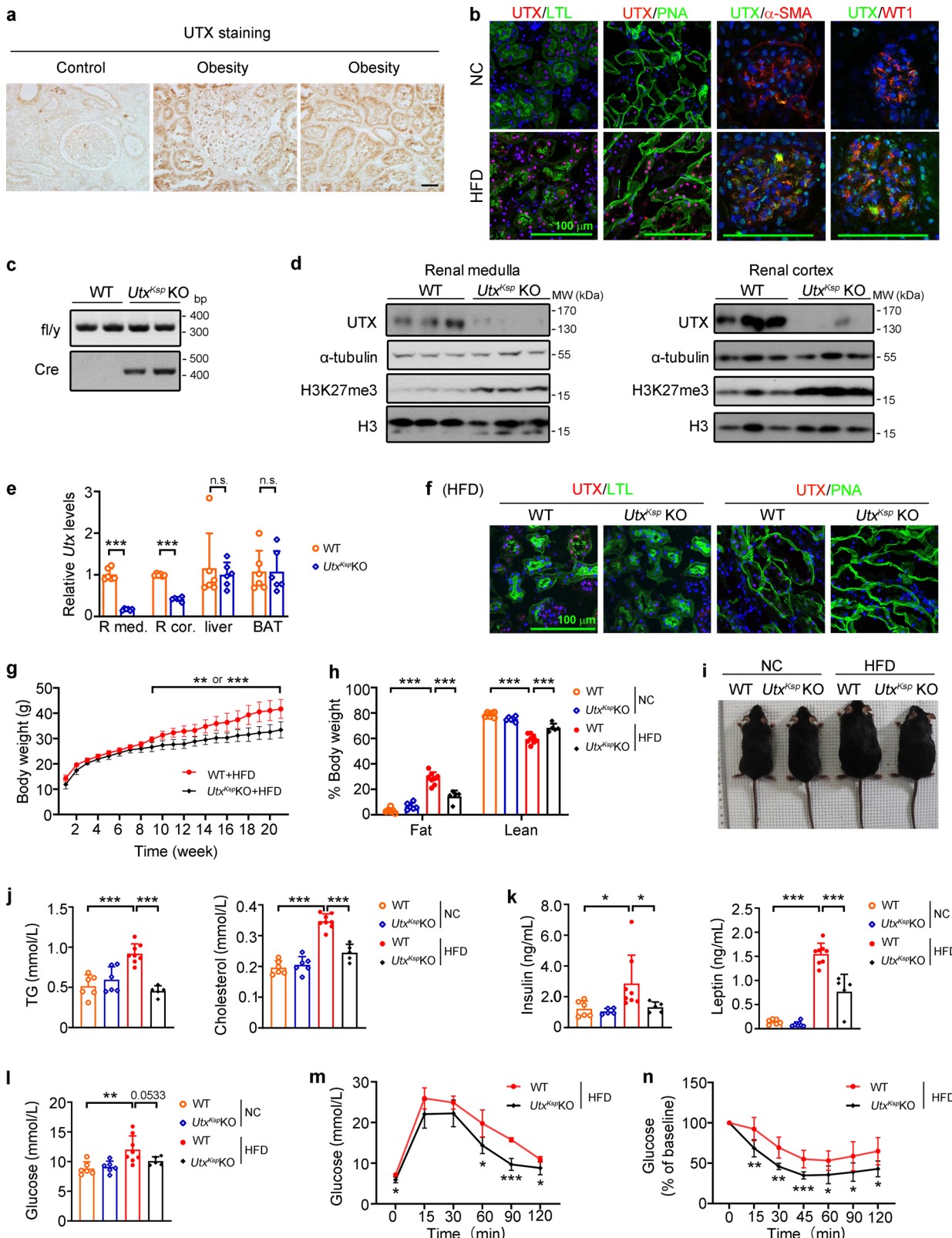

involved in TG synthesis and lipid storage, such as *Mogat1*, *Gpam*, *Cidea* and *Cidec*, in HFD-fed *Utx^Ksp* KO or *Utx^Pax2* KO mice, compared to their respective controls (Fig. 2c–f and Supplementary Fig. 3c). Downregulation of *Cidea* and *Cidec* by knockout *Utx* in kidney was found in NC-fed mice (Fig. 2e, f and Supplementary Fig. 3c). UTX has been reported to promote gene transcription through removing H3K27me3 from their

promoters. Increased H3K27me3 levels on the promoters of *Cidea* and *Cidec* were found upon; knockout as demonstrated by ChIP assay (Supplementary Fig. 3d). Moreover, the transcription levels of genes involved in lipid synthesis, lipid transport, β-oxidation, gluconeogenesis and glycolysis pathways, as well as protein levels of several key factors regulating these pathways were examined in mouse kidney. Under HFD stress, *Utx^Ksp* KO

**Fig. 1 $Utx^{Ksp}$ KO mouse was resistant to HFD-induced obesity. a** Representative images of renal UTX staining of human subjects, $n = 4$ or five independent subjects with control or obesity, respectively. Scale bar, 50 μm. Brown color indicates positive staining. **b** Representative images of UTX (red)/LTL (lotus tetragonolobus lectin, detecting proximal tubules; green), UTX (red)/PNA (peanut agglutinin, detecting distal tubules and collecting ducts; green), UTX (green)/α-SMA (alpha-smooth muscle actin, mesangial cell marker; red), and UTX (green)/WT1 (Wilms' tumor 1, podocyte marker; red) staining on the renal sections of WT ($n = 6$) and HFD ($n = 4$) mice. DAPI staining shown in blue. Scale bar, 100 μm. **c** Genotyping of WT and $Utx^{Ksp}$ KO mice. **d** Western blots of UTX, α-tubulin, H3K27me3 and H3 in renal medulla (left, $n = 4$) and cortex (right, $n = 6$) of WT and $Utx^{Ksp}$ KO mice. **e** qPCR results of $Utx$ in renal medulla (R med.), renal cortex (R cor.), liver and BAT of WT and $Utx^{Ksp}$ KO mice. $n = 6$ independent animals (mean ± SD); $^{***}P_{R\ med.} < 0.0001$, $^{***}P_{R\ cor.} < 0.0001$ (unpaired, two-tailed $t$-test). **f** Representative images of UTX (red)/LTL (green), UTX (red)/PNA (green) staining on the renal sections of HFD-fed WT and $Utx^{Ksp}$ KO mice. DAPI (blue) stained nuclei. Scale bar, 100 μm, $n = 4$ independent animals. **g–i**, Growth curves (**g**), body mass (**h**), and gross view (**i**) of WT and $Utx^{Ksp}$ KO mice. **g** $n = 8$, 5 independent animals for WT + HFD or $Utx^{Ksp}$ KO + HFD group, respectively (mean ± SD); $^{**}P_{body\ weight}$ or $^{***}P_{body\ weight}$ from 0.0035 to 0.0007; unpaired, two-tailed $t$-test; **h** $n = 6$, 6, 8, five independent animals for WT + NC or $Utx^{Ksp}$ KO + NC or WT + HFD or $Utx^{Ksp}$ KO + HFD group, respectively (mean ± SD); $^{***}P_{Fat\ (WT+NC\ vs\ WT+HFD)} < 0.0001$, $^{***}P_{Fat\ (WT+HFD\ vs\ Utx^{Ksp}\ KO+HFD)} < 0.0001$; $^{***}P_{Lean\ (WT+NC\ vs\ WT+HFD)} < 0.0001$, $^{***}P_{Lean\ (WT+HFD\ vs\ Utx^{Ksp}\ KO\ +HFD)} < 0.0001$ (one-way ANOVA). **j–l** Serum levels of TG (**j**, left), cholesterol (**j**, right), insulin (**k**, left), leptin (**k**, right) and non-fasting blood glucose (**l**) of WT and $Utx^{Ksp}$ KO mice, $n = 6$, 6, 8, 5 independent animals for WT + NC or $Utx^{Ksp}$ KO + NC or WT + HFD or $Utx^{Ksp}$ KO + HFD group, respectively (mean ± SD); $^{***}P_{TG\ (WT+NC\ vs\ WT+HFD)} < 0.0001$, $^{***}P_{TG\ (WT+HFD\ vs\ Utx^{Ksp}\ KO+HFD)} < 0.0001$; $^{***}P_{cholesterol\ (WT+NC\ vs\ WT+HFD)} < 0.0001$, $^{***}P_{cholesterol\ (WT+HFD\ vs\ Utx^{Ksp}\ KO+HFD)} < 0.0001$; $^{*}P_{Insulin\ (WT+NC\ vs\ WT+HFD)} = 0.0294$, $^{*}P_{Insulin\ (WT+HFD\ vs\ Utx^{Ksp}\ KO+HFD)} = 0.0378$; $^{***}P_{Leptin\ (WT+NC\ vs\ WT+HFD)} < 0.0001$, $^{***}P_{Leptin\ (WT+HFD\ vs\ Utx^{Ksp}\ KO+HFD)} < 0.0001$; $^{**}P_{glucose\ (WT+HFD\ vs\ WT+HFD)} = 0.0027$ (one-way ANOVA). **m, n** GTT (**m**) and ITT (**n**) results of WT and $Utx^{Ksp}$ KO mice. **m** $n = 4$, five independent animals for WT + HFD or $Utx^{Ksp}$ KO + HFD group, respectively (mean ± SD), $^{*}P_{GTT(0)} = 0.0214$; $^{*}P_{GTT(60)} = 0.0373$; $^{***}P_{GTT(90)} = 0.0003$, $^{*}P_{GTT(120)} = 0.0428$; (**n**) $n = 8$, five independent animals for WT + HFD or $Utx^{Ksp}$ KO + HFD group, respectively (mean ± SD), $^{**}P_{ITT(15)} = 0.0063$, $^{**}P_{ITT(30)} = 0.0011$, $^{***}P_{ITT(45)} = 0.0009$, $^{*}P_{ITT(60)} = 0.0238$, $^{*}P_{ITT(90)} = 0.0341$, $^{*}P_{ITT(120)} = 0.0125$, unpaired, two-tailed $t$-test. Source data are provided in the Source Data file.

resulted in decreased *Fasn*, *Glut4*, *G6pc* and *Pc* mRNA levels, with no obvious effect on other genes or proteins we examined (Supplementary Fig. 4).

**Kidney-specific knockout of *Utx* reduces hepatic steatosis and adipocyte size under HFD stress.** Anatomic analysis indicated significantly reduced liver weight in HFD-fed $Utx^{Ksp}$ KO or $Utx^{Pax2}$ KO mice (Supplementary Tables 2 and 3). Furthermore, ameliorated hepatic steatosis was found in dissected liver, and by H&E and Oil Red O staining, in HFD-fed $Utx^{Ksp}$ KO or $Utx^{Pax2}$ KO mice (Fig. 3a–c and Supplementary Fig. 5a). Furthermore, reduced hepatic TG and cholesterol levels, as well as down-regulated TG synthesis and storage related genes, such as *Mogat1*, *Cidea*, *Cidec*, and *Gpam* were also observed in the liver of HFD-fed $Utx^{Ksp}$ KO mice (Fig. 3d, e). Meanwhile, compared with HFD-fed WT mice, HFD-fed $Utx^{Ksp}$ KO or $Utx^{Pax2}$ KO mice showed significantly reduced adipocyte size in BAT, iWAT and eWAT (Fig. 4a–c and Supplementary Fig. 5b–d). Significantly upregulated thermogenic genes, including *Ucp1*, *Elovl6*, *Dio2*, *Cox5b* and *Prdm16*, were found in iWAT and BAT of HFD-fed $Utx^{Ksp}$ KO mice (Fig. 4d, e).

**Renal UTX negatively regulates PHGDH and serine levels under HFD stress.** By analyzing human kidney datasets for chronic kidney disease (CKD; GSE66494) or diabetic kidney disease (DKD; GSE96804), we examined gene alterations involved in de novo serine synthesis and *UTX* (Fig. 5a, b). As the rate limiting enzyme for de novo serine synthesis, the transcription level of phosphoglycerate dehydrogenase (*PHGDH*) was down-regulated in the kidneys of patients with DKD or CKD (Fig. 5a, b). Consistently, immunohistochemical staining demonstrated decreased PHGDH expression in the kidneys of subjects with obesity or DKD (Fig. 5c and Supplementary Tables 1, 4). Next, eGFR, as well as serum levels of serine and glycine, which are respectively the product of de novo serine synthesis and serine transformation product via one-carbon metabolism, were measured in the individuals with normal body weight ($n = 39$), overweight ($n = 23$), obesity ($n = 24$) and obesity-related renal dysfunction ($n = 12$) (Supplementary Table 5). Downregulated eGFR was found in subjects with overweight and obesity, and was further downregulated in subjects with obesity-related renal

dysfunction (Fig. 5d). Significantly lower serum serine level was found in the subjects with obesity-related renal dysfunction (Fig. 5e). Serum serine level was negatively correlated with CREA and BUN levels, and was positively correlated with eGFR in these clinical samples (Fig. 5f, g, Supplementary Fig. 6, and Supplementary Table 5). Similar glycine levels were found in these groups, and a weak negative correlation with cholesterol was observed (Supplementary Fig. 7 and Supplementary Table 5).

Next, we studied whether knockout of *Utx* in whole kidney or renal epithelial cells regulates serine and PHGDH levels. Down-regulated PHGDH level was found in the kidney of HFD-fed WT mice, while its protein levels, but not mRNA levels, were significantly higher in HFD-fed $Utx^{Ksp}$ KO or $Utx^{Pax2}$ KO mice (Fig. 5i, j and Supplementary Fig. 8). In HFD-fed WT mice, decreased serine and glycine levels in renal medulla, serum and urine were found; whereas in HFD-fed $Utx^{Ksp}$ KO mice, normal renal medulla and serum serine/glycine levels, as well as partly rescued urine glycine level, were observed (Fig. 5k–n). On the other hand, liver or adipose tissue specific knockout *Utx* in male mice showed no effect on PHGDH level, and no effect on circulating serine level was observed in liver specific *Utx* knockout mice (Supplementary Fig. 9).

Consistent with in vivo results, reduced lipid accumulation, as demonstrated by Oil red O staining and TG measurement, were found in UTX knockout or knockdown HK-2 cells under palmitic acid (PA) treatment-induced stress that mimics in vivo hyperlipidemia (Fig. 6a, b and Supplementary Fig. 10a–c). Furthermore, knockout or knockdown of UTX in HK-2 cells upregulated PHGDH under NC- and PA-stressed conditions (Fig. 6c and Supplementary Fig. 10d); whereas overexpression of UTX increased lipid accumulation, downregulated PHGDH and upregulated transcription levels of genes involved in TG synthesis/lipid storage upon PA stress (Fig. 6d–g). To explore whether the histone demethylase activity of UTX affects the transcription, plasmids expressing wildtype or a demethylase activity abolished UTX mutant were transfected in UTX knockout HK-2 cells. UTX knockout significantly decreased the mRNA levels of *CIDEA* and *CIDEC*, which were restored by over-expression of wildtype UTX but not the mutant (Supplementary Fig. 10e). Furthermore, an in vitro luciferase reporter assay confirmed the binding and regulation of UTX on *CIDEA* and *CIDEC* (Supplementary Fig. 10f).

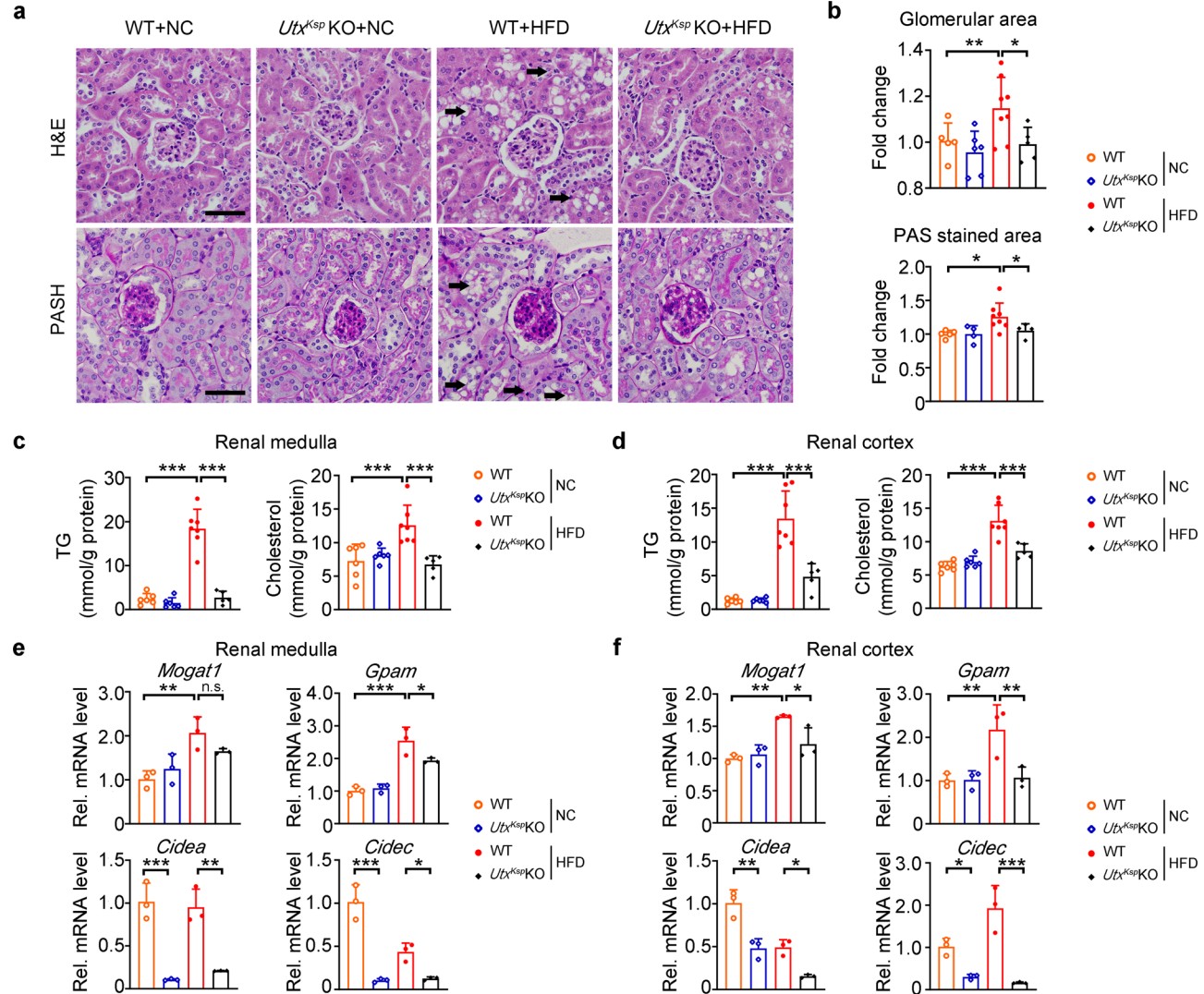

**Fig. 2 Reduced glomerular hypertrophy and renal lipid accumulation in *Utx^Ksp* KO mouse under HFD stress. a, b** H&E and PAS staining on renal sections of indicated groups (**a**), with quantitative data for relative glomerular area and relative mesanglial area (**b**), data shown as mean ± SD. $n_{H\&E}$ = 5, 6, 8, 5 independent animals for WT + NC or *Utx^Ksp* KO + NC or WT + HFD or *Utx^Ksp* KO + HFD group, respectively, $^{**}P_{H\&E\ (WT+NC\ vs\ WT+HFD)}$ = 0.0075, $^{*}P_{H\&E\ (WT+HFD\ vs\ Utx^{Ksp}\ KO+HFD)}$ = 0.0419; $n_{PAS}$ = 5, 4, 8, 4 independent animals for WT + NC or *Utx^Ksp* KO + NC or WT + HFD or *Utx^Ksp* KO + HFD group, respectively, $^{*}P_{PAS\ (WT+NC\ vs\ WT+HFD)}$ = 0.0211, $^{*}P_{PAS\ (WT+HFD\ vs\ Utx^{Ksp}\ KO+HFD)}$ = 0.0405 (one-way ANOVA). Scale bar, 50 μm. **c, d** TG (triglycerides) and cholesterol levels in renal medulla (**c**) and cortex (**d**) of the indicated groups; *n* = 6, 6, 7, 5 independent animals, for WT + NC or *Utx^Ksp* KO + NC or WT + HFD or *Utx^Ksp* KO + HFD group, respectively (mean ± SD). **c** $^{***}P_{TG\ (WT+NC\ vs\ WT+HFD)}$ < 0.0001, $^{***}P_{TG\ (WT+HFD\ vs\ Utx^{Ksp}\ KO+HFD)}$ < 0.0001; $^{***}P_{Cholesterol\ (WT+NC\ vs\ WT+HFD)}$ = 0.0009, $^{***}P_{Cholesterol\ (WT+HFD\ vs\ Utx^{Ksp}\ KO+HFD)}$ = 0.0006; **d** $^{***}P_{TG\ (WT+NC\ vs\ WT+HFD)}$ < 0.0001, $^{***}P_{TG\ (WT+HFD\ vs\ Utx^{Ksp}\ KO+HFD)}$ < 0.0001; $^{***}P_{Cholesterol\ (WT+NC\ vs\ WT+HFD)}$ < 0.0001, $^{***}P_{Cholesterol\ (WT+HFD\ vs\ Utx^{Ksp}\ KO+HFD)}$ = 0.0001 (one-way ANOVA). **e, f** Transcriptional levels of TG synthesis/lipid storage related genes in renal medulla (**e**) and cortex (**f**) of the indicated groups; *n* = 3 independent animals (mean ± SD). **e** $^{**}P_{Mogat1(WT+NC\ vs\ WT+HFD)}$ = 0.0033; $^{***}P_{Gpam\ (WT+NC\ vs\ WT+HFD)}$ = 0.0001, $^{*}P_{Gpam\ (WT+HFD\ vs\ Utx^{Ksp}\ KO+HFD)}$ = 0.0294; $^{***}P_{Cidea\ (WT+NC\ vs\ Utx^{Ksp}\ KO+NC)}$ = 0.0004, $^{**}P_{Cidea\ (WT+HFD\ vs\ Utx^{Ksp}\ KO+HFD)}$ = 0.0015; $^{***}P_{Cidec\ (WT+NC\ vs\ Utx^{Ksp}\ KO+NC)}$ < 0.0001, $^{*}P_{Cidec(WT+HFD\ vs\ Utx^{Ksp}\ KO+HFD)}$ = 0.0402; **f** $^{**}P_{Mogat1\ (WT+NC\ vs\ WT+HFD)}$ = 0.0018, $^{*}P_{Mogat1(WT+HFD\ vs\ Utx^{Ksp}\ KO+HFD)}$ = 0.0204; $^{**}P_{Gpam\ (WT+NC\ vs\ WT+HFD)}$ = 0.0072, $^{**}P_{Gpam\ (WT+HFD\ vs\ Utx^{Ksp}\ KO+HFD)}$ = 0.0097; $^{**}P_{Cidea\ (WT+NC\ vs\ Utx^{Ksp}\ KO+NC)}$ = 0.0013, $^{*}P_{Cidea\ (WT+HFD\ vs\ Utx^{Ksp}\ KO+HFD)}$ = 0.0188; $^{*}P_{Cidec\ (WT+NC\ vs\ Utx^{Ksp}\ KO+NC)}$ = 0.0279, $^{***}P_{Cidec\ (WT+HFD\ vs\ Utx^{Ksp}\ KO+HFD)}$ = 0.0003 (one-way ANOVA). Source data are provided in the Source Data file.

Since knockout or knockdown of UTX upregulated PHGDH level in HK-2 cells, we generated PHGDH knockout cells (Supplementary Fig. 10g). In PA-treated HK-2 cells, knockout of PHGDH increased lipid accumulation, transcription levels of genes involved in TG synthesis/storage and decreased intracellular serine/glycine levels (Fig. 6h–k). Consistent with PHGDH knockout results, in PA-treated HK-2 cells, PHGDH overexpression reduced lipid accumulation and transcription levels of genes involved in TG synthesis/storage (Supplementary Fig. 11).

To further investigate whether UTX-deficiency prevents PA-induced lipid accumulation via upregulating PHGDH, a stable UTX knockout and a UTX/PHGDH double knockout HK-2 cell (DKO) were generated (Supplementary Fig. 10h). Under PA stress, compared to UTX KO per se, DKO cells showed increased TG level and upregulated transcription levels of genes involved in TG synthesis/lipid storage (Fig. 7a, b). While knockout UTX normalized PA-induced reduction of intracellular serine and glycine levels, this normalization was abolished in DKO cells (Fig. 7c). Similar results were found for the medium serine level,

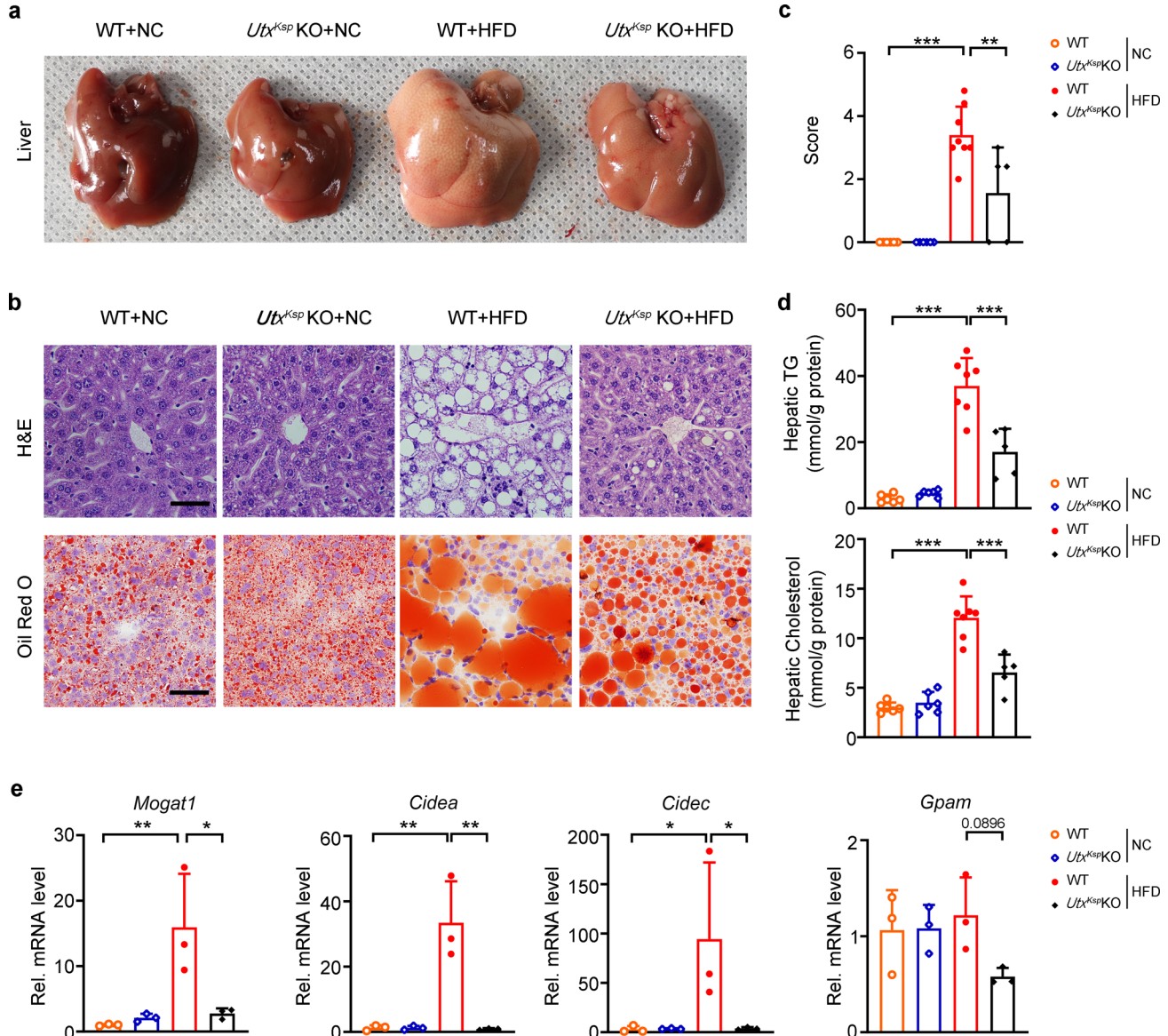

**Fig. 3 Reduced hepatic lipid accumulation and steatosis in *Utx*^Ksp^ KO mouse under HFD stress. a** Representative images of dissected liver of indicated groups. **b**, **c** Representative images of H&E and Oil Red O staining (**b**), with quantification for relative liver steatosis score (**c**) of the indicated groups, $n = 6$, 6, 8, 5 independent animals for WT + NC or *Utx*^Ksp^ KO + NC or WT + HFD or *Utx*^Ksp^ KO + HFD group, respectively (mean ± SD). $^{***}P_{(WT+NC\ vs\ WT+HFD)} < 0.0001$, $^{**}P_{(WT+HFD\ vs\ Utx^{Ksp}\ KO+HFD)} = 0.002$ (one-way ANONA). Scale bar, 50 μm. **d**, **e** Hepatic TG (triglycerides) and cholesterol levels (**d**) and TG synthesis/lipid storage related gene levels (**e**) of indicated groups. **d** $n = 6$, 6, 7, 5 independent animals for WT + NC or *Utx*^Ksp^ KO + NC or WT + HFD or *Utx*^Ksp^ KO + HFD group, respectively (mean ± SD), $^{***}P_{TG\ (WT+NC\ vs\ WT+HFD)} < 0.0001$, $^{***}P_{TG\ (WT+HFD\ vs\ Utx^{Ksp}\ KO+HFD)} < 0.0001$; $^{***}P_{Cholesterol\ (WT+NC\ vs\ WT+HFD)} < 0.0001$, $^{***}P_{Cholesterol\ (WT+HFD\ vs\ Utx^{Ksp}\ KO+HFD)} < 0.0001$; (**e**) $n = 3$ independent animals (mean ± SD), $^{**}P_{Mogat1\ (WT+NC\ vs\ WT+HFD)} = 0.0055$, $^{*}P_{Mogat1\ (WT+HFD\ vs\ Utx^{Ksp}\ KO+HFD)} = 0.0112$; $^{**}P_{Cidea\ (WT+NC\ vs\ WT+HFD)} = 0.0012$, $^{**}P_{Cidea\ (WT+HFD\ vs\ Utx^{Ksp}\ KO+HFD)} = 0.0011$; $^{*}P_{Cidec\ (WT+NC\ vs\ WT+HFD)} = 0.0498$, $^{*}P_{Cidec\ (WT+HFD\ vs\ Utx^{Ksp}\ KO+HFD)} = 0.0495$ (one-way ANOVA). Source data are provided in the Source Data file.

but not glycine level, in these groups (Fig. 7d). Next, we administrated NCT-503, a PHGDH inhibitor[24], to PA-treated UTX KO HK-2 cells. Significantly upregulated TG levels, transcriptional levels of genes involved in TG synthesis/lipid storage, as well as downregulated cellular/medium serine level and cellular glycine level, were observed (Fig. 7e–h). A control study indicated the specificity of NCT-503 on the enzymatic activity of PHGDH (Supplementary Fig. 12).

**UTX negatively regulates PHGDH by promoting ubiquitination at K310 and K330.** Ubiquitin-proteasome and autophagy mediated proteolysis are two major eukaryotic protein

degradation pathways[25]. To investigate how UTX negatively regulates PHGDH protein level in renal cells, we first determined the major PHGDH degradation pathway affected by UTX. Inhibitors for autophagosome (3-methyladenine, 3-MA), lysosome (chloroquine, CQ) and proteasome (MG132) were administrated to the control or UTX overexpressing HK-2 cells, only MG132 abolished the negative regulation of UTX on PHGDH levels (Fig. 8a), suggesting UTX proteasome-dependently mediates PHGDH degradation.

To identify possible binding partners of UTX, immunoprecipitation (IP) was performed in 293 T cells transfected with UTX, following mass spectrometromic analysis suggested that RNF114, an E3 ligase, may bind to UTX (Supplementary Fig. 13a). Under

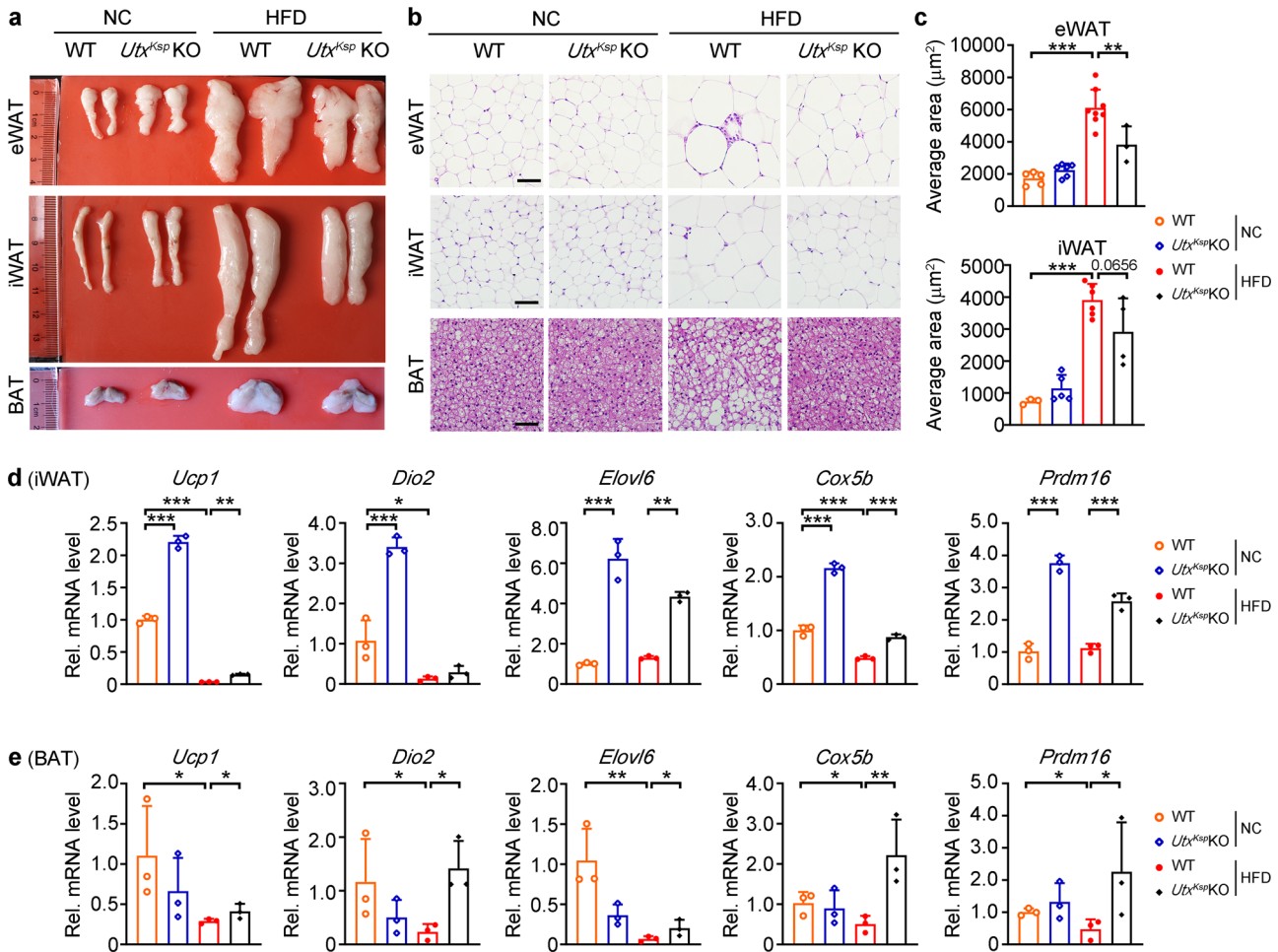

**Fig. 4 Decreased hypertrophy and increased thermogenesis in the adipose tissue of $Utx^{Ksp}$ KO mouse under HFD stress. a** Representative images of dissected adipose tissues of indicated groups, $n = 3$–8 independent animals. **b, c** H&E staining on different adipose tissues of indicated groups (**b**), with average cell area of eWAT and iWAT (**c**), data shown as mean ± SD. $n_{eWAT}$ = 5, 6, 8, 3 independent animals for WT + NC or $Utx^{Ksp}$ KO + NC or WT + HFD or $Utx^{Ksp}$ KO + HFD group, respectively, $***P_{eWAT\ (WT+NC\ vs\ WT+HFD)} < 0.0001$, $**P_{eWAT\ (WT+HFD\ vs\ Utx^{Ksp}\ KO+HFD)} = 0.0026$; $n_{iWAT}$ = 3, 5, 6, 4 independent animals for WT + NC or $Utx^{Ksp}$ KO + NC or WT + HFD or $Utx^{Ksp}$ KO + HFD group, respectively, $***P_{iWAT\ (WT+NC\ vs\ WT+HFD)} < 0.0001$ (one-way ANOVA). scale bar, 50 μm. **d, e** Transcriptional levels of thermogenic genes in iWAT (**d**) and BAT (**e**) of indicated groups; $n = 3$ independent animals (mean ± SD). **d** $***P_{Ucp1\ (WT+NC\ vs\ Utx^{Ksp}\ KO+NC)} < 0.0001$, $***P_{Ucp1\ (WT+NC\ vs\ WT+HFD)} < 0.0001$, $**P_{Ucp1\ (WT+HFD\ vs\ Utx^{Ksp}\ KO+HFD)} = 0.0028$; $***P_{Dio2\ (WT+NC\ vs\ Utx^{Ksp}\ KO+NC)} < 0.0001$, $*P_{Dio2\ (WT+NC\ vs\ WT+HFD)} = 0.0179$; $***P_{Elovl6\ (WT+NC\ vs\ Utx^{Ksp}\ KO+NC)} < 0.0001$, $**P_{Elovl6\ (WT+HFD\ vs\ Utx^{Ksp}\ KO+HFD)} = 0.0026$; $***P_{Cox5b\ (WT+NC\ vs\ Utx^{Ksp}\ KO+NC)} < 0.0001$, $***P_{Cox5b\ (WT+NC\ vs\ WT+HFD)} < 0.0001$, $***P_{Cox5b\ (WT+HFD\ vs\ Utx^{Ksp}\ KO+HFD)} = 0.0007$; $***P_{Prdm16\ (WT+NC\ vs\ Utx^{Ksp}\ KO+NC)} < 0.0001$, $***P_{Prdm16\ (WT+HFD\ vs\ Utx^{Ksp}\ KO+HFD)} = 0.0002$ (one-way ANOVA). **e** $*P_{Ucp1(WT+NC\ vs\ WT+HFD)} = 0.0495$, $*P_{Ucp1(WT+HFD\ vs\ Utx^{Ksp}\ KO+HFD)} = 0.0495$; $*P_{Dio2\ (WT+NC\ vs\ WT+HFD)} = 0.0495$, $*P_{Dio2(WT+HFD\ vs\ Utx^{Ksp}\ KO+HFD)} = 0.0495$; $**P_{Elovl6\ (WT+NC\ vs\ WT+HFD)} = 0.0015$, $*P_{Elovl6\ (WT+HFD\ vs\ Utx^{Ksp}\ KO+HFD)} = 0.0495$; $*P_{Prdm16\ (WT+NC\ vs\ WT+HFD)} = 0.0495$, $*P_{Prdm16\ (WT+HFD\ vs\ Utx^{Ksp}\ KO+HFD)} = 0.0495$; $*P_{Cox5b\ (WT+NC\ vs\ WT+HFD)} = 0.0495$, $**P_{Cox5b\ (WT+HFD\ vs\ Utx^{Ksp}\ KO+HFD)} = 0.0099$ (one-way ANOVA). iWAT inguinal white adipose tissue, eWAT epididymal adipose tissue, BAT brown adipose tissue. Source data are provided in the Source Data file.

overexpression conditions, co-IP experiments demonstrated binding of UTX with RNF114, and PHGDH binds to the UTX/ RNF114 complex (Fig. 8b–d and Supplementary Fig. 14a–c). Endogenous co-IP experiments indicated that PHGDH was weakly associated with UTX and RNF114 at the normal conditions, and their association was dramatically increased after PA stress, partly due to upregulated UTX and RNF114 expression (Fig. 8e). Increased RNF114 level was also found in the kidney of HFD-fed mice (Supplementary Fig. 13b). Knockout of UTX abolished the binding between endogenous PHGDH and RNF114 (Fig. 8f); moreover, overexpression of either UTX or RNF114 promoted the degradation of PHGDH in HK-2 cells (Fig. 8g). Since UTX upregulates the transcription of *CIDEA* and *CIDEC* (Supplementary Figs. 3c, d and 10e, f), while UTX is the only component in the PHGDH/UTX/RNF114 complex with known

direct effect on transcription, we explored whether PHGDH or RNF114 affects gene transcription by regulating UTX protein level. However, no change in UTX level was observed in PHGDH KO or RNF114 KO HK-2 cells (Supplementary Fig. 14d).

To address whether UTX regulates PHGDH stability by ubiquitination, we performed ubiquitination assays, and overexpression of UTX markedly increased ubiquitination of PHGDH (Fig. 8h), while UTX knockout inhibited ubiquitination of PHGDH (Fig. 8i). We next screened and identified that the K48-linked polyubiquitin chain was removed from PHGDH by the UTX/RNF114 complex (Fig. 8j). To determine the ubiquitination site(s), based on bioinformatic prediction (https://www.genecards.org) and a recent mass-spectrometry based PHGDH ubiquitination sites mapping study[24], we determined six lysines (K146, K289, K310, K330, K364, and

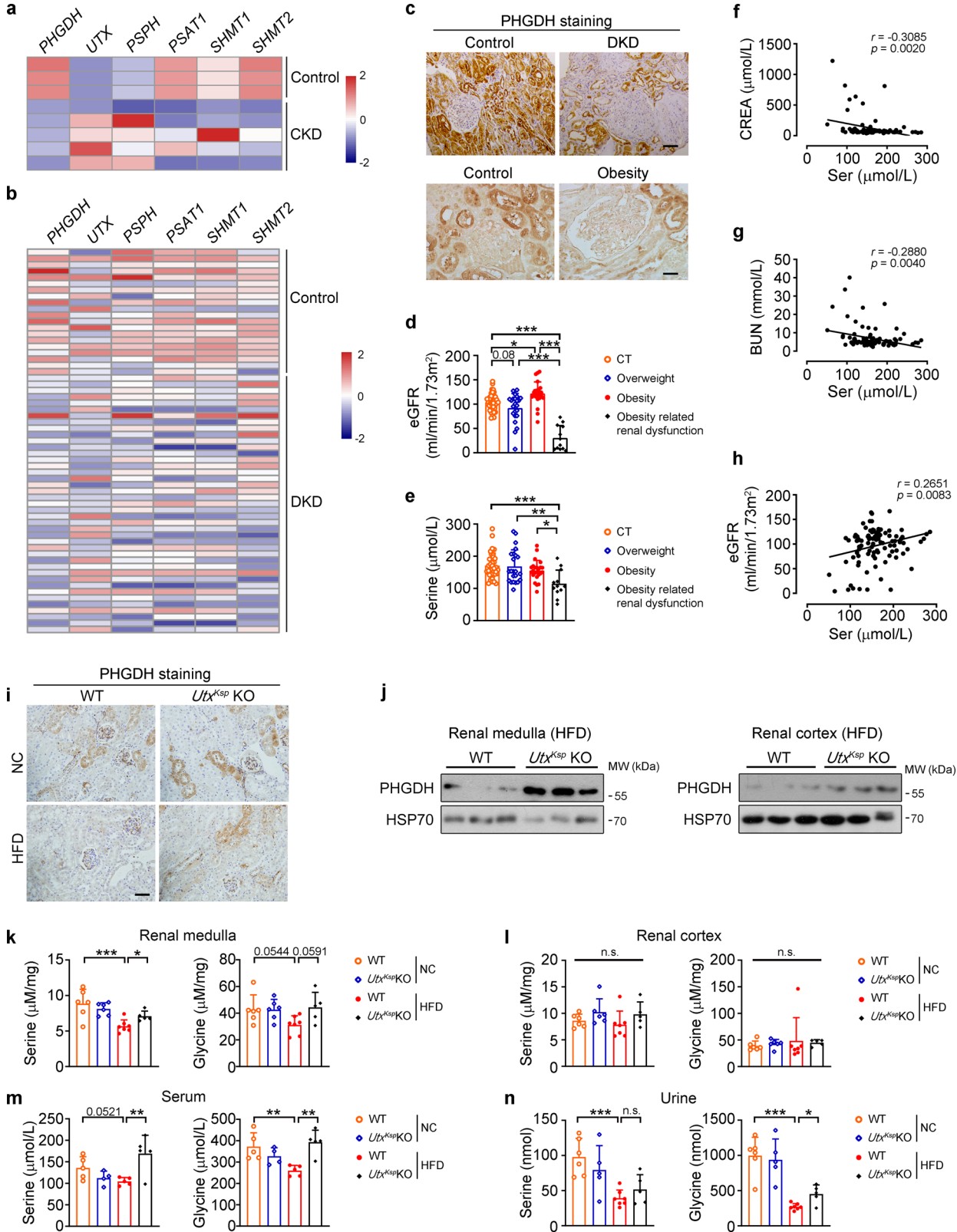

K384) as potential ubiquitination site(s). Six respective lysine to arginine PHGDH mutants were constructed. Compared to the wildtype, K310R and K330R mutations dramatically abolished UTX-promoted PHGDH ubiquitination (Fig. 8k), and markedly restored PHGDH level (Fig. 8l). These results indicate K310 and K330, which respectively reside on the surface of substrate binding domain and regulatory domain (Fig. 8m), are

sites for UTX/RNF114 mediated ubiquitination and degradation of PHGDH.

**Serine inhibits PA-induced lipid accumulation in cultured cells.** We further investigated whether serine, the product catalyzed by PHGDH, regulates lipid accumulation. Serine (200 or

**Fig. 5 Increased renal PHGDH and serine levels in $Utx^{Ksp}$ KO mice under HFD stress. a, b** Microarray data of de novo serine synthesis genes and *UTX* in the kidneys of patients with chronic kidney disease (CKD; **a**) or diabetic kidney disease (DKD; **b**). **c** Representative images of PHGDH staining in the kidneys of patients with DKD (top) and obesity (bottom); $n = $ 4-9; scale bar, 50 µm. Brown color indicates positive staining. **d–g** eGFR (**d**), serum serine level (**e**), correlation analysis between the serum serine level and serum CREA (creatinine, **f**) or serum BUN (blood urea nitrogen, **g**) level or eGFR (**h**) from healthy individuals ($n = 39$), overweight ($n = 23$), obesity ($n = 24$) and obesity-related renal disease ($n = 12$). **d, e** data shown as mean ± SD. $^{*}P_{eGFR\ (CT\ vs\ obesity)} = 0.0227$, $^{***}P_{eGFR\ (CT\ vs\ obesity\ related\ renal\ dysfunciton)} < 0.0001$, $^{***}P_{eGFR\ (overweight\ vs\ obesity\ related\ renal\ dysfunciton)} < 0.0001$, $^{***}P_{eGFR\ (obesity\ vs\ obesity\ related\ renal\ dysfunciton)} < 0.0001$; $^{***}P_{serine\ (CT\ vs\ obesity\ related\ renal\ dysfunciton)} = 0.0004$, $^{**}P_{serine\ (overweight\ vs\ obesity\ related\ renal\ dysfunciton)} = 0.0022$, $^{*}P_{serine\ (obesity\ vs\ obesity\ related\ renal\ dysfunciton)} = 0.0309$ (one-way ANOVA). **f–h** correlation analysis was performed by Pearson's method, the Pearson correlation coefficients and $p$ values (two-tailed test) were shown. **i** Representative images of renal PHGDH staining of WT and $Utx^{Ksp}$ KO mice under NC or HFD conditions. Scale bar, 50 µm. **j** Western blots of PHGDH and HSP70 levels in the renal medulla and cortex of indicated groups, $n = 3$ independent animals. **k–n** Serine and glycine levels in renal medulla (**k**), renal cortex (**l**), serum (**m**), and urine (**n**) of indicated groups, data shown as mean ± SD. **k–l** $n = 6, 6, 7, 5$ independent animals for WT + NC or $Utx^{Ksp}$ KO + NC or WT + HFD or $Utx^{Ksp}$ KO + HFD group, respectively; (**k**) $^{***}P_{serine\ (WT+NC\ vs\ WT+HFD)} = 0.0005$, $^{*}P_{serine\ (WT+HFD\ vs\ Utx^{Ksp}\ KO+HFD)} = 0.0126$; **m** $n = 5, 4, 5, 5$ independent animals for WT + NC or $Utx^{Ksp}$ KO + NC or WT + HFD or $Utx^{Ksp}$ KO + HFD group, respectively; $^{**}P_{serine\ (WT+HFD\ vs\ Utx^{Ksp}\ KO+HFD)} = 0.005$; $^{**}P_{glycine\ (WT+NC\ vs\ WT+HFD)} = 0.0068$, $^{**}P_{glycine\ (WT+HFD\ vs\ Utx^{Ksp}\ KO+HFD)} = 0.0018$; (**n**) $n = 6, 5, 7, 5$ independent animals for WT + NC or $Utx^{Ksp}$ KO + NC or WT + HFD or $Utx^{Ksp}$ KO + HFD group, respectively, $^{***}P_{serine\ (WT+NC\ vs\ WT+HFD)} = 0.0009$; $^{***}P_{glycine\ (WT+NC\ vs\ WT+HFD)} < 0.0001$, $^{*}P_{glycine\ (WT+HFD\ vs\ Utx^{Ksp}\ KO+HFD)} = 0.033$ (one-way ANOVA). Source data are provided in the Source Data file.

400 µM) significantly downregulated TG accumulation and genes involved in TG synthesis/storage in PA-treated HK-2 cells (Fig. 9a, b). These beneficial effects were significantly inhibited by knockout of serine transporter ASCT1 and/or ASCT2 (Fig. 9c–f). Moreover, serine treatment and PHGDH overexpression increased the NAD$^+$/NADH ratio in PA-treated HK-2 cells (Supplementary Fig. 15).

The attenuated hepatic steatosis found in HFD-fed $Utx^{Ksp}$ KO mice (Fig. 3) let us hypothesize that knockout of *Utx* in renal epithelial cells may result in releasing factors that affect liver lipid metabolism. To test this possibility, culture medium from differently treated HK-2 cells were collected and added to primary mouse hepatocytes (Supplementary Fig. 16a, d). Under PA stress, compared to cells treated with medium from respective control cells, reduced lipid accumulation was found in hepatic cells treated with medium from UTX KO cells; whereas increased lipid accumulation was found in those treated with medium from UTX overexpressing cells (Supplementary Fig. 16b, c, e, and f).

In UTX KO HK-2 cells, a larger amount of serine was released to the medium (Fig. 7d), we next tested whether exogenous serine regulates lipid level in hepatic cells. In PA-treated primary mouse hepatocytes, serine (200 or 400 µM) treatment also significantly downregulated TG accumulation and genes involved in TG synthesis/lipid storage (Fig. 9g, h). These effects were similarly observed in HepG2 cells and were inhibited by knockout of serine transporter ASCT1 and/or ASCT2 (Fig. 9i–l). Together, our results indicate that renal synthesized and released serine may inhibit lipid accumulation in hepatocytes. Similarly, glycine treatment also significantly decreased TG accumulation in PA-treated hepatocytes (Supplementary Fig. 17). Next, we examined whether serine affects adipocyte differentiation and lipid accumulation, and no obvious effect on adipogenesis and lipid accumulation were found upon serine treatment (Supplementary Fig. 18).

**Serine prevents HFD-induced renal dysfunction and hepatic steatosis.** Next, by supplementing serine in drinking water for 18 weeks (Fig. 10a), we investigated whether serine affects lipid accumulation in the kidney and liver of HFD-fed mice. At 8 weeks after HFD feeding, similar glucose tolerance and insulin tolerance were found in mice with or without serine treatment (Supplementary Fig. 19a, b). At 18 weeks after HFD feeding, increased serum serine and glycine levels, as well as reduced serum TG and cholesterol levels, were found after serine treatment (Fig. 10b, c). Decreased glomerular hypertrophy and lipid droplet formation in renal tubules (demonstrated by H&E staining), increased renal serine and glycine levels, decreased renal TG and cholesterol levels, downregulated genes involved in TG synthesis/storage, and similar insulin signaling related p-AKT level were found in the kidney of serine-treated HFD-fed mice (Fig. 10d–h, and Supplementary Fig. 19c, d). Similarly, decreased liver weights, hepatic steatosis (demonstrated by H&E staining and scoring), increased hepatic serine level, decreased hepatic TG and cholesterol levels, downregulated genes involved in TG synthesis/storage, as well as unchanged insulin signaling pathway, were found in the liver of serine-treated HFD-fed mice (Fig. 10i–m, Supplementary Fig. 19e, f and Supplementary Table 6). Whereas serine treatment showed no obvious effect on adipose tissue weights and adipose hypertrophy (Supplementary Table 6 and Supplementary Fig. 20).

## Discussion

In metabolic diseases, multiple organs, as well as their releasing factors, are simultaneously affected to generates an altered signaling network. Elucidating such tissue crosstalk is vital for understanding and preventing/treating diseases. Kidney, besides its well-known urinary function, also releases several circulating hormones, indicating its endocrinal role[12,13]. Kidney also releases some amino acids into the renal venous, for example, it is a major organ for serine synthesis[26,27]. Here, we demonstrated that renal synthesized serine, which is regulated by UTX, affects body metabolic homeostasis, especially the liver.

Increased UTX has been found in multiple renal cell types in diabetic patients and murine models by others and us[16,18]. This phenomenon was also observed in individuals with obesity and HFD-fed mice (Fig. 1a, b). Therefore, we first generated whole kidney *Utx* knockout mice by crossing $Utx^{f/f}$ with *Pax2*-cre. To further explore the role of UTX in renal tubular cells, we generated the renal tubular specific *Utx* knockout mouse by crossing $Utx^{f/f}$ with *Ksp*-cre, which has previously been reported to also express in renal proximal tubular cells[28–30]. Since the proximal tubular cells are highly involved in lipid and glucose metabolism in the kidney[31], therefore it will be interesting as a future study to use a renal proximal tubular specific Cre, such as SGLT2-cre, to generate a proximal tubular specific *Utx* knockout mouse, and compare its phenotype with that of $Utx^{Ksp/Pax2}$ KO mice.

Our study suggested that $Utx^{Ksp/Pax2}$ KO activated serine biosynthesis by promoting PHGDH level in the kidney. Moreover, renal *Utx* knockout not only ameliorated the HFD-induced kidney steatosis, but also prevented HFD-induced obesity and NAFLD due to increasing circulating serine level; renal-released serine prevented hepatocytes and renal tubular cells from lipid accumulation upon HFD stress (Figs. 1–3, 5, and 10 and Supplementary Figs. 1–3, 5). In contrast, liver or adipose tissue

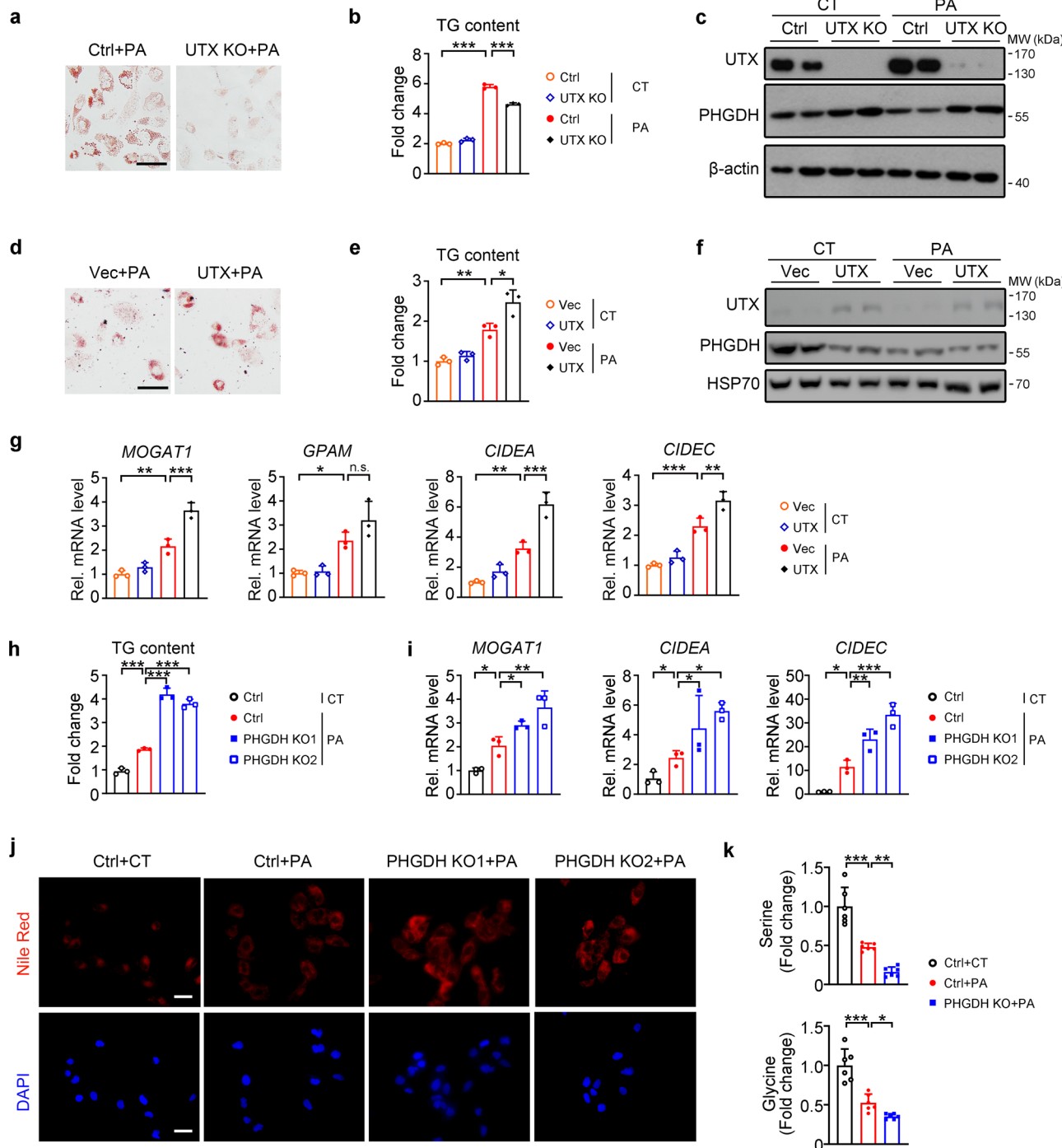

**Fig. 6 UTX and PHGDH regulated lipid level in cultured renal cells. a–c** UTX knockout prevented PA-induced lipid accumulation in HK-2 cells. Oil Red O staining (**a**), TG level (**b**) and PHGDH level (**c**) in UTX knockout HK-2 cells with or without PA stress. **b** $n = 3$ biological samples (mean ± SD), $^{***}P_{TG\ (Ctrl+CT\ vs\ Ctrl+PA)} < 0.0001$; $^{***}P_{TG\ (Ctrl+PA\ vs\ UTX\ KO+PA)} < 0.0001$ (one-way ANOVA); (**c**) $n = 2$ biological samples per group. **d–g** UTX overexpression exacerbated PA-induced lipid accumulation in HK-2 cells. Oil Red O staining (**d**), TG level (**e**), PHGDH level (**f**) and transcriptional levels of TG synthesis/lipid storage related genes (**g**) in UTX overexpressing HK-2 cells. **e**, **g** $n = 3$ biological samples (mean ± SD), $^{**}P_{TG\ (Vec+CT\ vs\ Vec+PA)} = 0.0063$, $^{*}P_{TG\ (Vec+PA\ vs\ UTX+PA)} = 0.0119$; $^{**}P_{MOGAT1(Vec+CT\ vs\ Vec+PA)} = 0.0033$, $^{***}P_{MOGAT1(Vec+PA\ vs\ UTX+PA)} = 0.0009$; $^{*}P_{GPAM\ (Vec+CT\ vs\ Vec+PA)} = 0.0293$; $^{**}P_{CIDEA\ (Vec+CT\ vs\ Vec+PA)} = 0.0038$, $^{***}P_{CIDEA\ (Vec+PA\ vs\ UTX+PA)} = 0.0009$; $^{***}P_{CIDEC\ (Vec+CT\ vs\ Vec+PA)} = 0.0009$, $^{**}P_{CIDEC\ (Vec+PA\ vs\ UTX+PA)} = 0.0076$ (one-way ANOVA); (**f**) $n = 2$ biological samples per group. **h–k** PHGDH knockout aggravated PA-induced lipid accumulation and reduced cellular serine/glycine levels in HK-2 cells. TG level (**h**), transcriptional levels of TG synthesis/storage related genes (**i**), Nile Red staining (**j**) and cellular serine (**k**, top) or glycine (**k**, bottom) level in PHGDH knockout HK-2 cells. Scale bar, 50 μm. **h–j** $n = 3$ biological samples per group, (**k**) $n = 6$ biological samples per group (mean ± SD). $^{***}P_{TG\ (Ctrl+CT\ vs\ Ctrl+PA)} = 0.0006$, $^{***}P_{TG\ (Ctrl+PA\ vs\ PHGDH\ KO1+PA)} < 0.0001$, $^{***}P_{TG\ (Ctrl+PA\ vs\ PHGDH\ KO2+PA)} < 0.0001$; $^{*}P_{MOGAT1\ (Ctrl+CT\ vs\ Ctrl+PA)} = 0.0351$, $^{*}P_{MOGAT1(Ctrl+PA\ vs\ PHGDH\ KO1+PA)} = 0.0408$, $^{**}P_{MOGAT1\ (Ctrl+PA\ vs\ PHGDH\ KO2+PA)} = 0.0034$; $^{*}P_{CIDEA(Ctrl+CT\ vs\ Ctrl+PA)} = 0.0230$, $^{*}P_{CIDEA\ (Ctrl+PA\ vs\ PHGDH\ KO1+PA)} = 0.0495$, $^{*}P_{CIDEA\ (Ctrl+PA\ vs\ PHGDH\ KO2+PA)} = 0.0289$; $^{*}P_{CIDEC\ (Ctrl+CT\ vs\ Ctrl+PA)} = 0.0158$, $^{**}P_{CIDEC\ (Ctrl+PA\ vs\ PHGDH\ KO1+PA)} = 0.0092$, $^{***}P_{CIDEC\ (Ctrl+PA\ vs\ PHGDH\ KO2+PA)} = 0.0002$; $^{***}P_{serine\ (Ctrl+CT\ vs\ Ctrl+PA)} < 0.0001$, $^{**}P_{serine\ (Ctrl+PA\ vs\ PHGDH\ KO+PA)} = 0.0036$; $^{***}P_{glycine\ (Ctrl+CT\ vs\ Ctrl+PA)} < 0.0001$, $^{*}P_{glycine\ (Ctrl+PA\ vs\ PHGDH\ KO+PA)} = 0.0101$ (one-way ANOVA). **a–k** at least three independent experiments were performed and similar results were obtained. Source data are provided in the Source Data file.

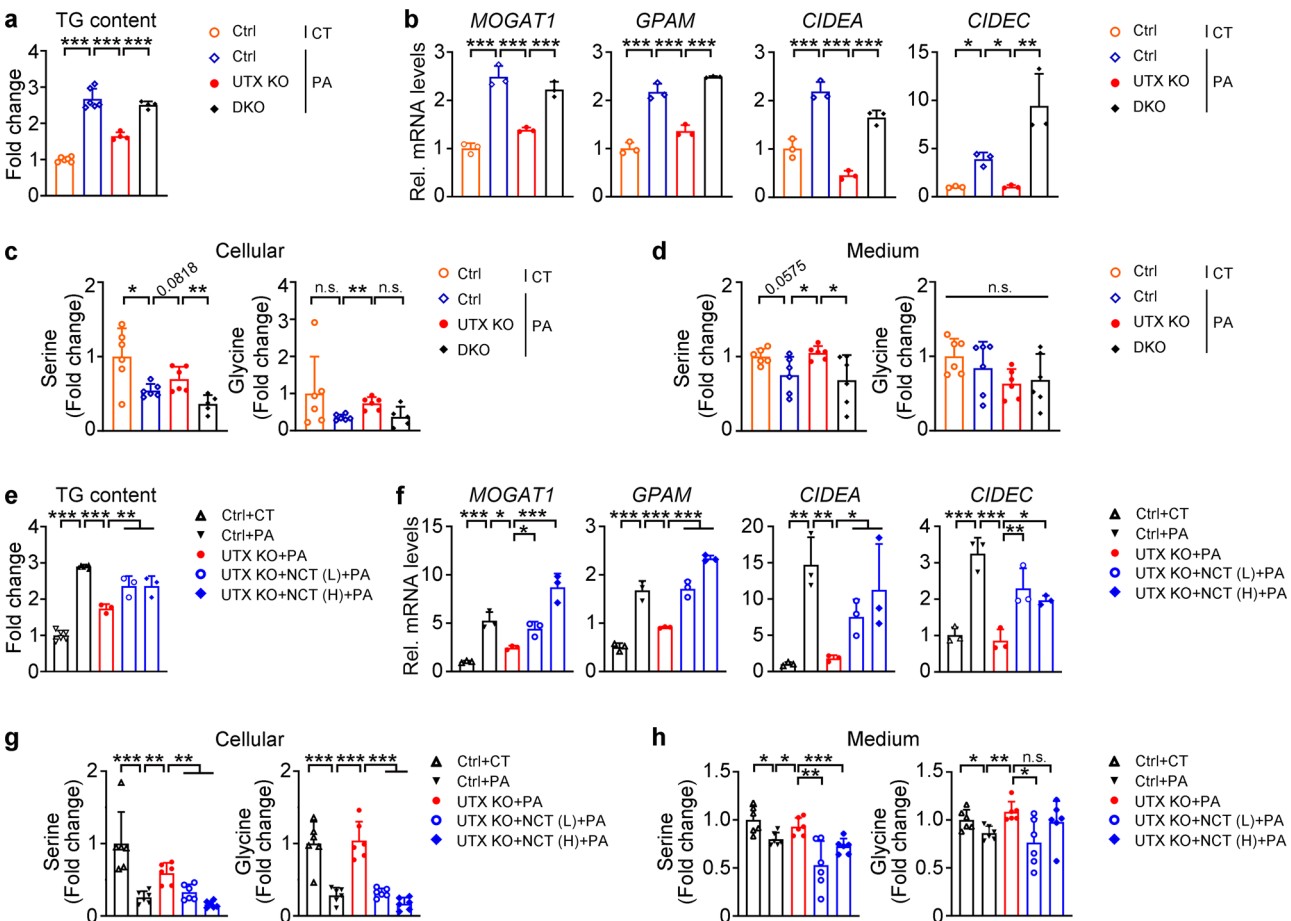

**Fig. 7 UTX regulated lipid level through enzymatic activity of PHGDH in cultured renal cells. a, b** TG level (**a**) and transcriptional levels of TG synthesis/ lipid storage related genes (**b**) in UTX knockout or UTX/PHGDH double knockout (DKO) HK-2 cells, **a** $n = 6, 6, 4, 4$ biological samples for Ctrl + CT or Ctrl+PA or UTX KO + PA or DKO + PA group, respectively (mean ± SD); **b** $n = 3$ biological samples per group (mean ± SD). ***$P_{TG\ (Ctrl+CT\ vs\ Ctrl+PA)} < 0.0001$, ***$P_{TG\ (Ctrl+PA\ vs\ UTX\ KO+PA)} < 0.0001$, ***$P_{TG\ (UTX\ KO+PA\ vs\ DKO+PA)} < 0.0001$; ***$P_{MOGAT1\ (Ctrl+CT\ vs\ Ctrl+PA)} < 0.0001$, ***$P_{MOGAT1\ (Ctrl+PA\ vs\ UTX\ KO+PA)} < 0.0001$, ***$P_{MOGAT1\ (UTX\ KO+PA\ vs\ DKO+PA)} = 0.0006$; ***$P_{GPAM\ (Ctrl+CT\ vs\ Ctrl+PA)} < 0.0001$, ***$P_{GPAM\ (Ctrl+PA\ vs\ UTX\ KO+PA)} = 0.0002$, ***$P_{GPAM\ (UTX\ KO+PA\ vs\ DKO+PA)} < 0.0001$; ***$P_{CIDEA\ (Ctrl+CT\ vs\ Ctrl+PA)} < 0.0001$, ***$P_{CIDEA\ (Ctrl+PA\ vs\ UTX\ KO+PA)} < 0.0001$, ***$P_{CIDEA\ (UTX\ KO+PA\ vs\ DKO+PA)} < 0.0001$; *$P_{CIDEC\ (Ctrl+CT\ vs\ Ctrl+PA)} = 0.0163$, *$P_{CIDEC\ (Ctrl+PA\ vs\ UTX\ KO+PA)} = 0.014$, **$P_{CIDEC\ (UTX\ KO+PA\ vs\ DKO+PA)} = 0.0014$ (one-way ANOVA). **c, d** Cellular (**c**) and medium (**d**) serine (left) or glycine (right) level in UTX knockout or UTX/PHGDH double knockout HK-2 cells, $n = 6$ biological samples per group (mean ± SD). **c** *$P_{serine\ (Ctrl+CT\ vs\ Ctrl+PA)} = 0.0109$, **$P_{serine\ (UTX\ KO+PA\ vs\ DKO+PA)} = 0.0039$; **$P_{glycine\ (Ctrl+PA\ vs\ UTX\ KO+PA)} = 0.0014$; (**d**) *$P_{serine\ (Ctrl+PA\ vs\ UTX\ KO+PA)} = 0.0299$, *$P_{serine\ (UTX\ KO+PA\ vs\ DKO+PA)} = 0.0415$ (one-way ANOVA). **e, f** TG level (**e**) and transcriptional levels of TG synthesis/lipid storage related genes (**f**) in NCT-503 treated UTX knockout HK-2 cells, (**e**) $n = 6, 6, 3, 3$ biological samples for Ctrl+CT or Ctrl+PA or UTX KO + PA or UTX KO + NCT (L) + PA or UTX KO + NCT (H) + PA group, respectively (mean ± SD). ***$P_{TG\ (Ctrl+CT\ vs\ Ctrl+PA)} < 0.0001$, ***$P_{TG\ (Ctrl+PA\ vs\ UTX\ KO+PA)} < 0.0001$, **$P_{TG\ (UTX\ KO+PA\ vs\ UTX\ KO+NCT\ (L)+PA)} = 0.0023$, **$P_{TG\ (UTX\ KO+PA\ vs\ UTX\ KO+NCT\ (H)+PA)} = 0.0023$; (**f**) $n = 3$ biological samples per group (mean ± SD) ***$P_{MOGAT1\ (Ctrl+CT\ vs\ Ctrl+PA)} = 0.0006$, *$P_{MOGAT1\ (Ctrl+PA\ vs\ UTX\ KO+PA)} = 0.0135$, *$P_{MOGAT1\ (UTX\ KO+PA\ vs\ UTX\ KO+NCT\ (L)+PA)} = 0.0348$, ***$P_{MOGAT1\ (UTX\ KO+PA\ vs\ UTX\ KO+NCT\ (H)+PA)} < 0.0001$; ***$P_{GPAM\ (Ctrl+CT\ vs\ Ctrl+PA)} < 0.0001$, ***$P_{GPAM\ (Ctrl+PA\ vs\ UTX\ KO+PA)} = 0.0001$, ***$P_{GPAM\ (UTX\ KO+PA\ vs\ UTX\ KO+NCT\ (L)+PA)} < 0.0001$, ***$P_{GPAM\ (UTX\ KO+PA\ vs\ UTX\ KO+NCT\ (H)+PA)} < 0.0001$; **$P_{CIDEA\ (Ctrl+CT\ vs\ Ctrl+PA)} = 0.0049$, **$P_{CIDEA\ (Ctrl+PA\ vs\ UTX\ KO+PA)} = 0.0073$, *$P_{CIDEA\ (UTX\ KO+PA\ vs\ UTX\ KO+NCT\ (L)+PA)} = 0.0495$, *$P_{CIDEA\ (UTX\ KO+PA\ vs\ UTX\ KO+NCT\ (H)+PA)} = 0.0470$; ***$P_{CIDEC\ (Ctrl+CT\ vs\ Ctrl+PA)} = 0.0001$, ***$P_{CIDEC\ (Ctrl+PA\ vs\ UTX\ KO+PA)} < 0.0001$, **$P_{CIDEC\ (UTX\ KO+PA\ vs\ UTX\ KO+NCT\ (L)+PA)} = 0.0046$, *$P_{CIDEC\ (UTX\ KO+PA\ vs\ UTX\ KO+NCT\ (H)+PA)} = 0.0243$ (one-way ANOVA). **g, h** Cellular (**g**) and medium (**h**) serine (left) or glycine (right) level in NCT-503 treated UTX knockout HK-2 cells, $n = 6$ biological samples per group (mean ± SD). **g** ***$P_{serine\ (Ctrl+CT\ vs\ Ctrl+PA)} < 0.0001$, **$P_{serine\ (Ctrl+PA\ vs\ UTX\ KO+PA)} = 0.001$, **$P_{serine\ (UTX\ KO+PA\ vs\ UTX\ KO+NCT\ (L)+PA)} = 0.005$, **$P_{serine\ (UTX\ KO+PA\ vs\ UTX\ KO+NCT\ (H)+PA)} = 0.002$; ***$P_{glycine\ (Ctrl+CT\ vs\ Ctrl+PA)} < 0.0001$, ***$P_{glycine\ (Ctrl+PA\ vs\ UTX\ KO+PA)} < 0.0001$, ***$P_{glycine\ (UTX\ KO+PA\ vs\ UTX\ KO+NCT\ (L)+PA)} < 0.0001$, ***$P_{glycine\ (UTX\ KO+PA\ vs\ UTX\ KO+NCT\ (H)+PA)} < 0.0001$; (**h**) *$P_{serine\ (Ctrl+CT\ vs\ Ctrl+PA)} = 0.0153$, *$P_{serine\ (Ctrl+PA\ vs\ UTX\ KO+PA)} = 0.021$, **$P_{serine\ (UTX\ KO+PA\ vs\ UTX\ KO+NCT\ (L)+PA)} = 0.009$, ***$P_{serine\ (UTX\ KO+PA\ vs\ UTX\ KO+NCT\ (H)+PA)} = 0.0005$; *$P_{glycine\ (Ctrl+CT\ vs\ Ctrl+PA)} = 0.0273$, **$P_{glycine\ (Ctrl+PA\ vs\ UTX\ KO+PA)} = 0.0022$, *$P_{glycine\ (UTX\ KO+PA\ vs\ UTX\ KO+NCT\ (L)+PA)} = 0.0188$ (one-way ANOVA). **a–h** At least three independent experiments were performed and similar results were obtained. Source data are provided in the Source Data file.

specific knockout *Utx* in male mice, showed no effect on circulation serine and PHGDH levels (Supplementary Figs. 2j, k and 9), which may due to the much lower RNF114 level in the liver and adipose tissues compared to that in the kidney (https://www.proteinatlas.org/ENSG00000124226-RNF114/tissue). Interestingly, renal knockout *Utx* also increased intestinal probiotics in HFD-fed mice (Supplementary Fig. 21). There are some

differences observed between *Utx^{Ksp}* KO and *Utx^{Pax2}* KO mice, such as the mildly increased fat weight in *Utx^{Ksp}* KO mice *vs.* no obvious change in fat weight in *Utx^{Pax2}* KO mice under NC conditions; and dramatically decreased fat weight in HFD-fed *Utx^{Ksp}* KO or *Utx^{Pax2}* KO mice (Fig. 1g, h, Supplementary Figs. 1h and 2a–c, Supplementary Tables 2 and 3). Such difference may be caused by the varied renal cell types knockout of *Utx*

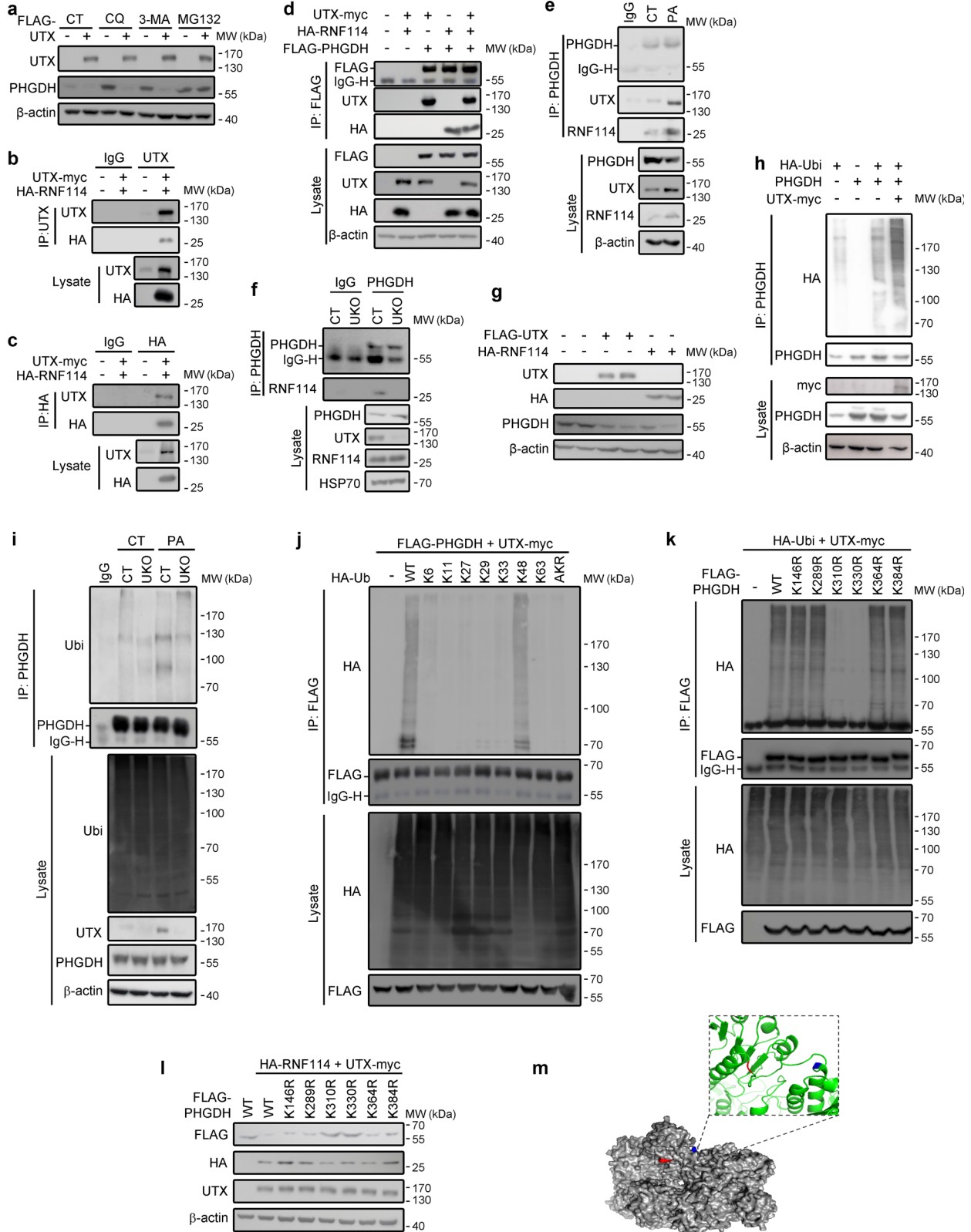

using different Cre strains, similar phenomena have been previously observed in other genes knockout mediated by different Cre strains in the same tissue[32,33].

Important functions of serine have attracted attention from different angles. Neuroprotective role of serine has been revealed[34,35], impaired serine biosynthesis has been found in Alzheimer's disease (AD)[36], therefore, serine is now being investigated as a therapy for neurological disorders including polyneuropathy, amyotrophic lateral sclerosis and AD[34,36,37]. As a critical one-carbon unit donor involved in methionine cycle and folate cycle which contribute to nucleotide synthesis, cancer cells utilize serine as the major source of one-carbon units to accelerate

**Fig. 8 UTX regulated PHGDH degradation in cultured cells. a** Effects of CQ, 3-MA and MG132 on UTX-mediated destabilization of PHGDH in HK-2 cells. **b**, **c** Co-immunoprecipitation by anti-UTX (**b**) or anti-HA (**c**) in HEK293T cells transfected with the indicated plasmids. **d** Co-immunoprecipitation of UTX, RNF114 and PHGDH in HEK293T cells transfected with indicated plasmids. β-actin serves as the loading control. **e** Endogenous co-immunoprecipitation of UTX, RNF114 and PHGDH with or without PA treatment in HK-2 cells. β-actin serves as the loading control. **f** Co-immunoprecipitation of endogenous PHGDH and RNF114 by anti-PHGDH in UTX KO or control HK-2 cells. HSP70 serves as the loading control. **g** Representative Western blots of UTX, RNF114 (HA), and PHGDH in HK-2 cells transfected with indicated plasmids. **h** Immunoblot analysis of the anti-PHGDH immunoprecipitates from HEK293T cells transfected with indicated plasmids and treated with MG132. **i** UTX knockout reduced ubiquitination of endogenous PHGDH in the HK-2 cells treated with MG132. β-actin serves as the loading control. **j**, **k** Immunoblot analysis of the anti-FLAG immunoprecipitate from HEK293T cells transfected with indicated plasmids and treated with MG132. K6, K11, K27, K29, K33, K48 and K63, ubiquitin mutants with respective lysine substituted by arginine; AKR, a ubiquitin mutant with all seven indicated lysines substituted by arginine; K146R, K289R, K310R, K330R, K364R, K384R, PHGDH mutants with respective lysine substituted by arginine. **l** Representative Western blots of UTX, HA, and FLAG in HK-2 cells transfected with indicated plasmids. **m** Crystal structure of PHGDH built by homology modeling. **a–l** Each experiment was repeated for at least three times with representative results shown. Source data are provided in the Source Data file.

cell growth and proliferation[38]. Serine also plays an important role in metabolic diseases. Significantly decreased serum serine level is found in patients with diabetes, CKD or NAFLD, as well as in ob/ob and db/db mice[26,27,39–41], indicating that serine may play an important role in metabolic diseases. In the present study, decreased serine level was also found in obesity related kidney disease, while renal knockout *Utx* increased circulating serine level (Figs. 5e and 5m). Whether eGFR affects circulating serine level is unclear. There was no significant change in eGFR between the WT and *Utx*^Ksp KO mice under NC- and HFD-fed conditions (Supplementary Fig. 1n), which indicates no correlation in our experimental setting; whereas a mild positive correlation between eGFR and circulating serine level was suggested in collected clinical samples (Fig. 5h). Therefore, it will be interesting to explore in a larger cohort with different renal disease groups.

Previous studies have indicated lipotoxicity as a major mechanism involve in HFD-induced glomerular hypertrophy[42]. Here, we found that serine treatment inhibited lipid accumulation in the tubular cells which may benefit glomerular hypertrophy (Fig. 10d, e). Meanwhile, glomerular hypertrophy in obesity-related nephropathy is associated with tubule-glomerular feedback induced glomerular hyperfiltration[43,44]; therefore, altered glomerular hypertrophy by serine treatment (Fig. 10d, e) may also due to tubule-glomerular feedback, similar to the effects of SGLT2 inhibitors[45], which will be interesting for future investigation.

Additionally, as a rate limiting enzyme of serine synthesis, PHGDH also play important roles in metabolic diseases and tumors[46,47]. For example, decreased hepatic PHGDH level has been found in the HFD- or MCD-induced steatosis animals, and in patients of alcoholic and non-alcoholic liver diseases, which is associated with lower hepatic serine levels[48]. Furthermore, *Phgdh* transgenic mice show significantly reduced hepatic triglyceride accumulation and lipogenic gene levels fed on HFD diet[48], indicating regulation of hepatic lipid level by PHGDH. However, specific KO of *Phgdh* mice in adipose tissues show no effect on body weight and glucose intolerance on males, but a mild reduced body weight and glucose intolerance on females upon a high-fat high sucrose diet[49], indicating tissue specific and gender specific effects of PHGDH.

Although the biological function of UTX in regulating gene transcription has been intensively studied[15,50], its function in protein stability is unknown. Accumulating evidence has suggested that histone-modification enzymes may regulate the ubiquitination of target proteins[51–53]. Here, we report an important role of UTX on the regulation of PHGDH stability, through interacting with a E3 ligase RNF114, the UTX/RNF114 complex ubiquitinates PHGDH at K310 and K330 and promotes its degradation (Fig. 8). Consistently, in vitro and in vivo results all suggested that stabilization of PHGDH and the resulting increase of serine synthesis contribute to the protective effects on renal

lipid accumulation after *Utx* knockout. These results thus reveal an important mechanism underlying the regulatory role of UTX in metabolic diseases. Both our results (Supplementary Fig. S13b) and a single-cell sequencing study suggested upregulated RNF114 under obesity or diabetic conditions in the kidney[54], therefore, upregulated expression of UTX and RNF114 in the kidney may contribute to the enhanced degradation of PHGDH. Furthermore, enhanced binding between UTX and RNF114 was found under PA stress (Fig. 8e). Since UTX forms a complex with RNF114 that leads to PHGDH degradation, thus enhanced binding may also induce increased PHGDH degradation. However, other factors that trigger the changes in HFD conditions still await further investigation.

In the present study, we showed that kidney-specific knockout UTX affected lipid accumulation in the liver and adipose tissue. Consistently, primary mouse hepatocytes and stromal vascular fraction (SVF) cells cultured with medium from differently treated HK-2 cells suggested that under PA stress, reduced lipid accumulation was found in hepatocytes and SVF cells treated with medium from UTX KO cells (Supplementary Figs. 16, 22). Renal UTX knockout increased PHGDH and circulation serine levels, while serine treatment also regulated lipid accumulation in the liver and primary mouse hepatocytes (Figs. 5, 9–10), but not affected in vitro adipogenesis and in vivo adipose hypertrophy under HFD stress (Supplementary Figs. 18, 20). Together, these results suggested that renal knockout UTX increased serine level which attenuated hepatic steatosis. On the other hand, renal knockout UTX also released unknown factor(s) that affect adipose tissue, however, serine was not the one. The effect of renal knockout UTX on adipose tissue warrants future investigations.

In summary, our findings reveal a renal UTX-PHGDH-serine axis regulates metabolic homeostasis in the kidney and liver (Fig. 10n). Dietary serine supplement may be further explored as a preventive/therapeutic approach against high-fat diets induced metabolic disorders.

## Methods

**Study approval**. Human serum samples were collected by Affiliated Wuhan Hospital of Traditional Chinese and Western Medicine of Tongji Medical School (Supplementary Table 5). Normal (CT), overweight or subjects with obesity were defined as BMI < 23 kg/m², 23 ≤ BMI < 27.5 kg/m², and BMI ≥ 27.5 kg/m², respectively as described[55,56]. Paraffin embedded human renal biopsy samples, including diabetic kidney disease samples or subjects with obesity, were collected by the Affiliated Union Hospital and Hubei Cancer Hospital of Tongji Medical College (Supplementary Tables 1, 4). Collection and use of patient samples and patient characteristics were approved by the Institutional Review Boards of affiliated Wuhan Hospital of Traditional Chinese and Western Medicine, Hubei Cancer Hospital and Union Hospital of Tongji Medical College (approval numbers (2020) 16, LLHBCH2022YN-015, and (2018)S196, respectively), all subjects included in the study provided written informed consent. Mice were handled according to the Guidelines of the China Animal Welfare Legislation, as approved by the Committee on Ethics in the Care and Use of Laboratory Animals of College of Life Sciences, Wuhan University.

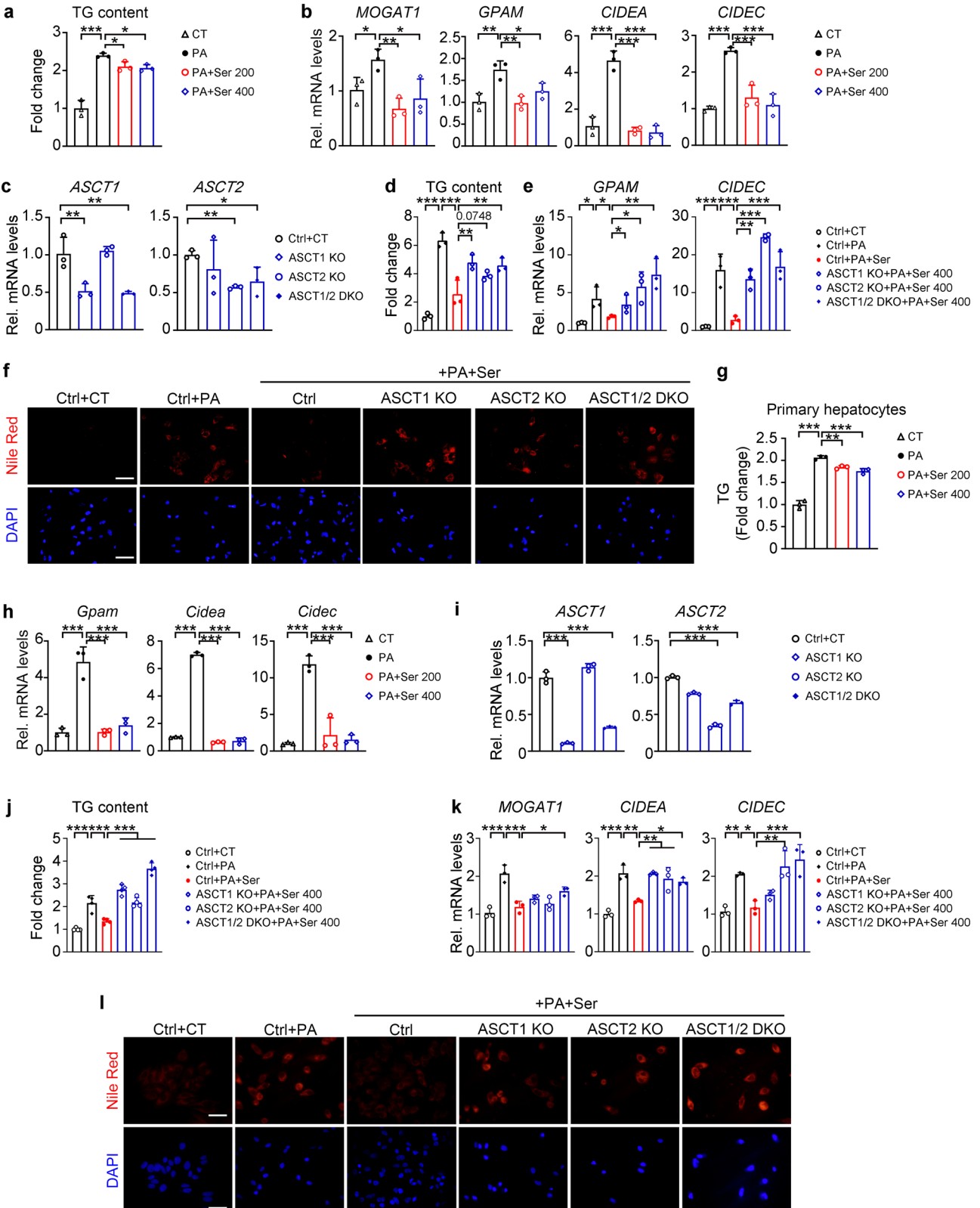

**Animals**. Mouse strains, including *Utx*<sup>flox</sup> **Animals**. Mouse strains, including $Utx^{flox}$ mice (JAX stock No.021926), *Pax2*-Cre mice (Cre activity expressing in glomeruli, renal tubules, collecting ducts, and nephric duct in the kidney[28]) and *Ksp*-Cre mice (CDH16-Cre, Cre activity expressing in renal tubules[29]) were used. Male *Pax2*-Cre or *Ksp*-Cre mice were crossed with $Utx^{fl/+}$ females to generate whole kidney specific or renal epithelial cell specific *Utx* knockout mice, respectively. Additionally, male *Adiponectin*-Cre (*Adi*-Cre, JAX, stock No.010803) or *Alb*-Cre (JAX, stock No.003574) mice were crossed with $Utx^{fl/+}$ females to generate liver or adipose tissue specific *Utx*

knockout mice. Genotyping was performed (primers listed in Supplementary Table 7). As an X-linked gene, males genotyped as $Utx^{fl/y}$ (WT) and $Utx^{fl/y}Pax2$-Cre ($Utx^{Pax2}$ KO), $Utx^{fl/y}Ksp$-Cre ($Utx^{Ksp}$ KO), $Utx^{fl/y}Adi$-Cre ($Utx^{Adi}$ KO) or $Utx^{fl/y}Alb$-Cre ($Utx^{Alb}$ KO) were used in the present study.

Mice were maintained in a specific-pathogen-free, temperature controlled ($22 \pm 1$ °C) animal facility with a 12-h light/dark cycle, and free access to water/food. Male mice were used in this study, euthanasia was performed by $CO_2$ inhalation. Mice were maintained on normal chow (NC, #1025, Beijing Huafukang,

**Fig. 9 Serine suppressed palmitic acid-induced renal and hepatic lipid accumulation through its transporter in cultured cells. a, b** Serine treatment prevented PA-induced lipid accumulation in HK-2 cells. TG level (**a**) and transcriptional levels of TG synthesis/lipid storage related genes (**b**) in serine-treated HK-2 cells under PA stress, $n = 3$ biological samples per group (mean ± SD). $^{***}P_{TG\ (CT\ vs\ PA)} < 0.0001$, $^{*}P_{TG\ (PA\ vs\ PA+Ser200)} = 0.0419$, $^{*}P_{TG\ (PA\ vs\ PA+Ser400)} = 0.0415$; $^{*}P_{MOGAT1\ (CT\ vs\ PA)} = 0.0342$, $^{**}P_{MOGAT1\ (PA\ vs\ PA+Ser200)} = 0.0061$, $^{*}P_{MOGAT1\ (PA\ vs\ PA+Ser400)} = 0.0214$; $^{**}P_{GPAM\ (CT\ vs\ PA)} = 0.0032$, $^{**}P_{GPAM\ (PA\ vs\ PA+Ser200)} = 0.0025$, $^{*}P_{GPAM\ (PA\ vs\ PA+Ser400)} = 0.0284$; $^{***}P_{CIDEA\ (CT\ vs\ PA)} < 0.0001$, $^{***}P_{CIDEA\ (PA\ vs\ PA+Ser200)} < 0.0001$, $^{***}P_{CIDEA\ (PA\ vs\ PA+Ser400)} < 0.0001$; $^{***}P_{CIDE\ (CT\ vs\ PA)} < 0.0001$, $^{***}P_{CIDEC\ (PA\ vs\ PA+Ser200)} = 0.0004$; $^{***}P_{CIDEC\ (PA\ vs\ PA+Ser400)} = 0.0002$ (one-way ANOVA). **c–f,** *ASCT1* and/or *ASCT2* knockout inhibited serine effects on PA-treated HK-2 cells. qPCR results of *ASCT1* and *ASCT2* levels (**c**), TG level (**d**), transcriptional levels of TG synthesis/lipid storage related genes (**e**) and Nile Red staining (**f**) in indicated HK-2 cells, $n = 3$ biological samples per group (mean ± SD). $^{**}P_{ASCT1\ (Ctrl+CT\ vs\ ASCT1\ KO)} = 0.0031$, $^{**}P_{ASCT1\ (Ctrl+CT\ vs\ ASCT1/2\ DKO)} = 0.0023$; $^{**}P_{ASCT2\ (Ctrl+CT\ vs\ ASCT2\ KO)} = 0.0024$, $^{*}P_{ASCT2\ (Ctrl+CT\ vs\ ASCT1/2\ DKO)} = 0.0495$; $^{***}P_{TG\ (Ctrl+CT\ vs\ Ctrl+PA)} < 0.0001$, $^{***}P_{TG\ (Ctrl+PA\ vs\ Ctrl+PA+Ser)} < 0.0001$, $^{**}P_{TG\ (Ctrl+PA+Ser\ vs\ ASCT1\ KO+PA+Ser400)} = 0.0048$, $^{**}P_{TG\ (Ctrl+PA+Ser\ vs\ ASCT1/2\ DKO+PA+Ser400)} = 0.0098$; $^{*}P_{GPAM\ (Ctrl+CT\ vs\ Ctrl+PA)} = 0.0495$, $^{*}P_{GPAM\ (Ctrl+PA\ vs\ Ctrl+PA+Ser)} = 0.0495$, $^{*}P_{GPAM\ (Ctrl+PA+Ser\ vs\ ASCT1\ KO+PA+Ser400)} = 0.0495$, $^{*}P_{GPAM\ (Ctrl+PA+Ser\ vs\ ASCT2\ KO+PA+Ser400)} = 0.0495$, $^{**}P_{GPAM\ (Ctrl+PA+Ser\ vs\ ASCT1/2\ DKO+PA+Ser400)} = 0.0053$; $^{***}P_{CIDEC\ (Ctrl+CT\ vs\ Ctrl+PA)} = 0.0002$, $^{***}P_{CIDEC\ (Ctrl+PA\ vs\ Ctrl+PA+Ser)} = 0.0006$, $^{**}P_{CIDEC\ (Ctrl+PA+Ser\ vs\ ASCT1\ KO+PA+Ser400)} = 0.0033$, $^{***}P_{CIDEC\ (Ctrl+PA+Ser\ vs\ ASCT2\ KO+PA+Ser400)} < 0.0001$, $^{***}P_{CIDEC\ (Ctrl+PA+Ser\ vs\ ASCT1/2\ DKO+PA+Ser400)} = 0.0003$ (one-way ANOVA). **g, h** Serine treatment prevented PA-induced lipid accumulation in primary mouse hepatocytes. TG level (**g**) and transcription levels of TG synthesis/lipid storage related genes (**h**) in primary mouse hepatocytes treated with serine under PA stress, $n = 3$ biological samples per group (mean ± SD). **g** $^{***}P_{TG\ (CT\ vs\ PA)} < 0.0001$, $^{**}P_{TG\ (PA\ vs\ PA+Ser200)} = 0.0057$, $^{***}P_{TG\ (PA\ vs\ PA+Ser400)} = 0.0007$; $^{***}P_{Gpam\ (CT\ vs\ PA)} < 0.0001$, $^{***}P_{Gpam\ (PA\ vs\ PA+Ser200)} < 0.0001$, $^{***}P_{Gpam\ (PA\ vs\ PA+Ser400)} < 0.0001$; $^{***}P_{Cidea\ (CT\ vs\ PA)} < 0.0001$, $^{***}P_{Cidea\ (PA\ vs\ PA+Ser200)} < 0.0001$, $^{***}P_{Cidea\ (PA\ vs\ PA+Ser400)} < 0.0001$; $^{***}P_{Cidec\ (CT\ vs\ PA)} < 0.0001$, $^{***}P_{Cidec\ (PA\ vs\ PA+Ser200)} < 0.0001$, $^{***}P_{Cidec\ (PA\ vs\ PA+Ser400)} < 0.0001$ (one-way ANOVA). **i–l** *ASCT1* and/or *ASCT2* knockout inhibited serine effects on PA-treated HepG2 cells. qPCR results of *ASCT1* and *ASCT2* levels (**i**), TG level (**j**), transcriptional levels of TG synthesis/lipid storage related genes (**k**) and Nile Red staining (**l**) in indicated HepG2 cells, $n = 3$ biological samples per group (mean ± SD). $^{***}P_{ASCT1(Ctrl\ vs\ ASCT1\ KO)} < 0.0001$, $^{***}P_{ASCT1\ (Ctrl\ vs\ ASCT1/2\ DKO)} < 0.0001$; $^{***}P_{ASCT2\ (Ctrl\ vs\ ASCT2\ KO)} < 0.0001$, $^{***}P_{ASCT2\ (Ctrl\ vs\ ASCT1/2\ DKO)} < 0.0001$; $^{***}P_{TG\ (Ctrl+CT\ vs\ Ctrl+PA)} < 0.0001$, $^{***}P_{TG\ (Ctrl+PA\ vs\ Ctrl+PA+Ser)} = 0.0008$, $^{***}P_{TG\ (Ctrl+PA+Ser\ vs\ ASCT1\ KO+PA+Ser400)} < 0.0001$, $^{***}P_{TG\ (Ctrl+PA+Ser\ vs\ ASCT2\ KO+PA+Ser400)} = 0.0005$, $^{***}P_{TG\ (Ctrl+PA+Ser\ vs\ ASCT1/2\ DKO+PA+Ser400)} < 0.0001$; $^{***}P_{MOGAT1\ (Ctrl+CT\ vs\ Ctrl+PA)} < 0.0001$, $^{***}P_{MOGAT1\ (Ctrl+PA\ vs\ Ctrl+PA+Ser)} = 0.0002$, $^{*}P_{MOGAT1\ (Ctrl+PA+Ser\ vs\ ASCT1/2\ DKO+PA+Ser400)} = 0.0242$; $^{***}P_{CIDEA\ (Ctrl+CT\ vs\ Ctrl+PA)} < 0.0001$, $^{**}P_{CIDEA\ (Ctrl+PA\ vs\ Ctrl+PA+Ser)} = 0.0013$, $^{**}P_{CIDEA\ (Ctrl+PA+Ser\ vs\ ASCT1\ KO+PA+Ser400)} = 0.0014$, $^{**}P_{CIDEA\ (Ctrl+PA+Ser\ vs\ ASCT2\ KO+PA+Ser400)} = 0.0076$, $^{*}P_{CIDEA\ (Ctrl+PA+Ser\ vs\ ASCT1/2\ DKO+PA+Ser400)} = 0.0213$; $^{**}P_{CIDEC\ (Ctrl+CT\ vs\ Ctrl+PA)} = 0.005$, $^{*}P_{CIDEC\ (Ctrl+PA\ vs\ Ctrl+PA+Ser)} = 0.0116$, $^{**}P_{CIDEC\ (Ctrl+PA+Ser\ vs\ ASCT2\ KO+PA+Ser400)} = 0.0023$, $^{***}P_{CIDEC\ (Ctrl+PA+Ser\ vs\ ASCT1/2\ DKO+PA+Ser400)} = 0.0006$ (one-way ANOVA). **a–l** At least three independent experiments were performed and similar results were obtained. Source data are provided in the Source Data file.

Beijing, China). For diet-induced obesity-related kidney disease, mice were fed with a high-fat diet (60% Kcal fat, #D12492, Research Diets, New Brunswick, NJ) at 4 weeks-old for indicated period of time. For serine treatments, L-serine (Sigma, St. Louis, MO) was supplemented in drinking water (5%, w/v) for indicated period of time.

**Biochemical measurements.** Serum levels of leptin, insulin, triglyceride and cholesterol were measured using mouse leptin ELISA kit (Millipore, Billerica, MA), rat/mouse insulin ELISA kit (Millipore), triglyceride kit (Jiancheng, Nanjing, China), and total cholesterol kit (Kehua Biotech, Shanghai, China), respectively[57,58]. Triglyceride and cholesterol levels in the kidney or liver were measured using triglyceride kit (Jiancheng) and total cholesterol kit (Kehua Biotech), respectively. Glucose tolerance test (GTT) and insulin tolerance test (ITT) were performed[59]. For GTT, mice were fasted overnight, then intraperitoneally injected with D-glucose (1.5 g/kg body weight; Amresco, Solon, OH). For ITT, mice were fasted for 6 h, then intraperitoneally injected with insulin (0.75 unit/kg body weight; Lily, France). Blood glucose levels were measured with a OneTouch blood glucose meter (LifeScan, Milpitas, CA) at 0, 15, 30, 60, 90 and 120 min after glucose or insulin injection. Body composition were scanned with a minispec LF-50 analyzer (Bruker, Rheinstetten, Germany). Twenty-four hour urine samples from WT and $Utx^{Ksp}$ KO mice, fed on NC and HFD diets, were collected in metabolic cages (Tecniplast, Italy) 1 day before sacrifice, and the volume was measured. Serum creatinine (CREA) level was analyzed on a Siemens ADVIA 2400 automatic biochemical analyzer using a creatinine reagent kit, while urine creatinine, albumin and BUN were measured with an Olympus AU2700 automatic biochemical analyzer using creatinine reagent kit, albumin reagent kit and urine BUN reagent kit, respectively (all from Fuxing Changzheng Medical, Shanghai, China). Mice eGFR was calculated by the creatinine clearance with the formula Ccr = CREA (urine)*Volume (urine)/ CREA (serum)/1440 (24 h)[60,61]. Cellular $NAD^+$ and NADH levels were determined using a $NAD^+$/NADH kit (Beyotime, Suzhou, China).

**Histological and immunohistochemical studies.** Paraffin embedded kidney, liver and adipose samples were sectioned and stained with hematoxylin & eosin (H&E) or periodic acid schiff-hematoxylin (PASH), analyses for glomerular area, PAS positive stained area and adipocyte size were performed[57,62].

Paraffin-embedded sections were deparaffinized and rehydrated as previously reported[63]. Sections were incubated with 3% $H_2O_2$ for 5 min to quench endogenous peroxidase activity. After blocking with 2% goat serum, primary antibodies including PHGDH (1:2000 dilution, Proteintech, Wuhan, China) was applied to the sections overnight at 4 °C. After washing with PBST, sections were incubated with biotinylated anti-rabbit (Vector laboratories, Burlingame, CA) for 1 h at room temperature. Positive staining was visualized using DAB substrate

following the protocol (Vector laboratories). Sample pictures were taken using an Olympus BX60 microscope with a digital camera.

For co-immunofluorescent staining, sections were incubated overnight with respective primary antibodies (Supplementary Table 8). For renal tubular staining, sections were incubated with 1 µg/mL PNA (peanut agglutinin, detecting distal tubules and collecting ducts), or 1 µg/mL LTL (lotus tetragonolobus lectin, detecting proximal tubules) (all from Vector Laboratories). While, α-SMA or WT1 was used to detect renal mesanglial cells or podocytes, respectively. After washing, sections were incubated with respective Alexa Fluor secondary antibody (Thermo Fisher, Waltham, MA). Sections were covered with DAPI and anti-fading medium. Images were taken by a confocal microscope (Leica SP8, Germany).

**Cell culture.** Human renal tubular cell line HK-2 (GDC0152, from CCTCC, China Center for Type Culture Collection, Wuhan, China) was cultured in DMEM/F12 media (Hyclone, Palo Alto, CA) plus 10% FBS. A human hepatocellular carcinoma cell line HepG2 (CL-0103000, from Procell Biotech, Wuhan, China) and HEK293T cells (CL-0005, Procell Biotech) were cultured in DMEM media (Hyclone) containing 10% FBS. HK-2 cells, HepG2 and HEK293T were analyzed with authenticated STR locus and tested for mycoplasma contamination, by CCTCC or Procell Biotech.

Primary mouse hepatocytes were isolated as described[64]. Briefly, mouse liver was perfused with calcium-free solution, digested with collagenase (Sigma), dispersed hepatocytes were collected and plated in collagen-coated plates with DMEM media plus 10% FBS. Plates were washed 4 h later and the adherent hepatocytes were used for further experiments. For medium treatment, medium was freshly collected and filtered from HK-2 cells treated with/without 400 µM palmitic acid (PA), then mixed with equal volume of fresh DMEM medium containing 10% FBS, and applied to 100% confluent primary hepatocytes or HepG2 cells for additional 48 h[65].

Mouse stromal vascular fraction (SVF) was isolated as reported[57]. Briefly, minced iWAT was digested at 37 °C in digestion buffer containing collagenase D and dispase II (Sigma), then filtered. Mature adipocytes and SVF cells were separated by centrifugation. Mouse SVFs were grown in DMEM/F12 media (Hyclone) supplemented with 10% FBS. Differentiation of primary white adipocytes was performed as described[57,66]. Briefly, confluent SVFs (Day 0) were induced by differentiation media (containing 0.5 mM IBMX, 1 µM dexamethasone and 10 µg/mL insulin) for 4 days, then changed to media containing 10 µg/mL insulin until Day 6.

**Plasmids, transfection, treatments.** Plasmids for human full length UTX (pFLAG-UTX), and shUTX were kind gifts from Dr. Min Gyu Lee (MD Anderson Cancer Center). CRISPR-Cas9 vectors containing gRNA targeting UTX (5'-ACCG AAAAGCGAGCGGCGAGAGCG-3'), PHGDH (5'-ACCGGCTCTGAGCCTCC

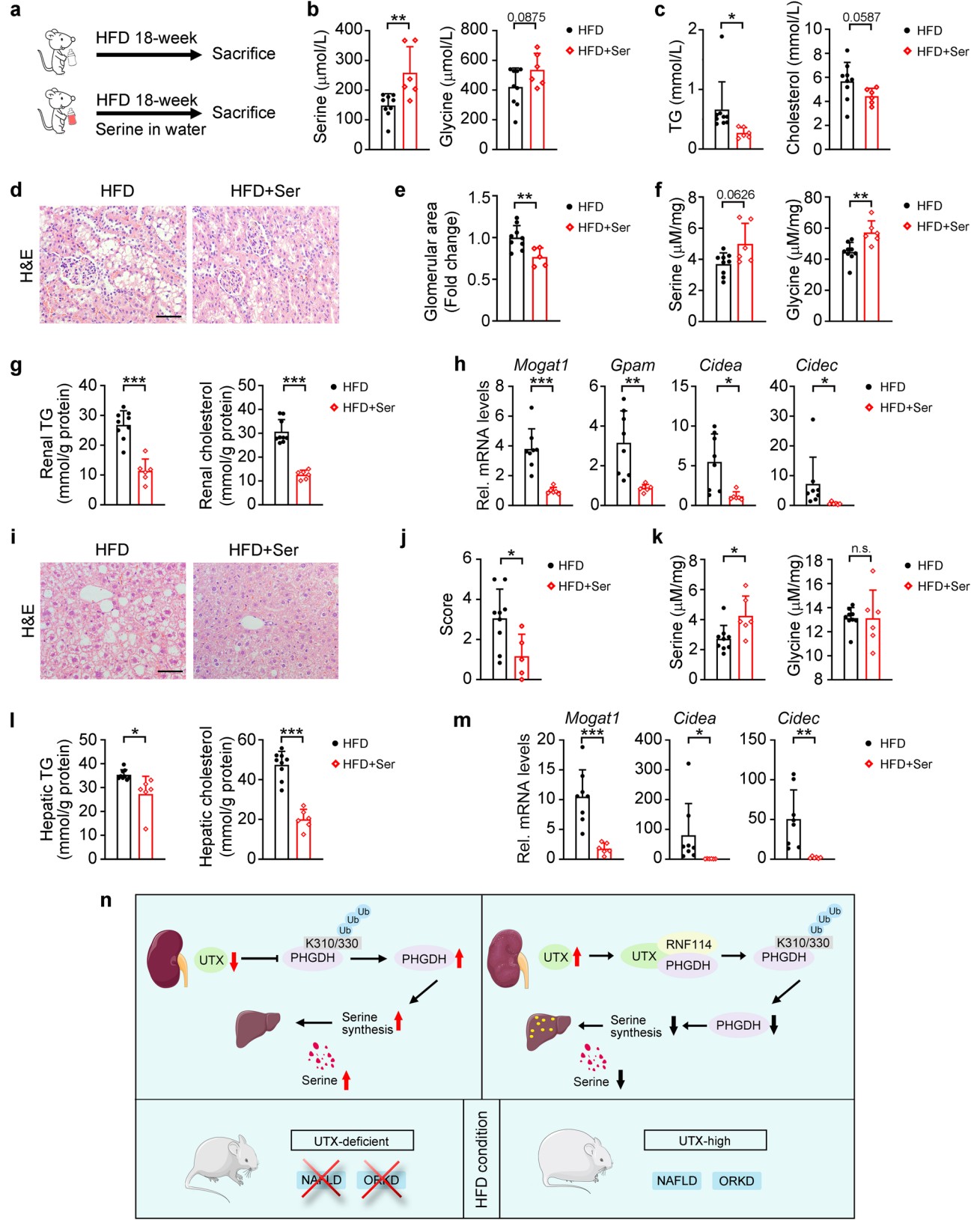

TTGGTGC-3' or 5'-ACCGAAAGCAGAACCTTAGCAAAG-3'), RNF114 (5'-CA
CCGGCCGCTTACACGTGTCCGCA-3'), ASCT1 (5'-CACCGTTGCAGCT
TTCCGTACGGTA-3'), ASCT2 (5'-CACCGCAGCGCCACACCAAAGACGA-3'),
were used to knockout UTX, PHGDH, RNF114 or ASCT1/2, respectively.
Expression plasmids for HA-tagged Ubi (WT, K6, K11, K27, K29, K33, K48, K63,
AKR) and myc-tagged UTX were constructed. HA-tagged RNF114, Flag-tagged
wildtype PHGDH and seven lysine to arginine mutants, pGL3-basic-CIDEA,

pGL3-basic-CIDEC plasmids were constructed with standard procedures (the
primers used are provided in Supplementary Table 9).

Most cells were transfected with indicated plasmids for 6 h, then treated with/
without 400 μM PA for 48 h before collection. Some cells were transfected with
indicated plasmids for 24 h, and treated with different inhibitors (CQ, chloroquine,
0.05 mM; 3-MA, 3-methyladenine, 3.35 mM; MG132, 10 μM) for 6 h. For serine
treatments, normally cultured or PA-stressed HK-2 cells, primary mouse

**Fig. 10 Serine suppressed renal and hepatic lipid accumulation in HFD-fed mice. a** Experiment design for serine administration via drinking water. **b, c** Concentrations of serine, glycine, TG and cholesterol in the serum of indicated groups, $n = 9$, 6 independent animals for HFD or HFD + Ser group, respectively (mean ± SD). $^{**}P_{serine} = 0.0053$; $^{*}P_{TG} = 0.0363$ (unpaired, two-tailed $t$-test). **d–h** H&E staining (**d**) with quantitative data for relative glomerular area (**e**), serine and glycine concentration (**f**), renal TG and cholesterol levels (**g**), and transcriptional levels of TG synthesis/lipid storage related genes (**h**) in the kidney of indicated groups, data shown as mean ± SD. **e** $n = 9$, 5 independent animals for HFD or HFD + Ser group, respectively, $^{**}P_{H\&E} = 0.0059$; **f, g** $n = 9$, 6 independent animals for HFD or HFD + Ser group, respectively, $^{**}P_{glycine} = 0.0073$; $^{***}P_{TG} < 0.0001$, $^{***}P_{Cholesterol} < 0.0001$; (**h**) $n = 8$, 6 independent animals for HFD or HFD + Ser group, respectively, $^{***}P_{Mogat1} = 0.0005$; $^{**}P_{Gpam} = 0.0049$; $^{*}P_{Cidea} = 0.01$; $^{*}P_{Cidec} = 0.0495$ (unpaired, two-tailed $t$-test). **i–m** H&E staining (**i**), with quantitative data for relative liver steatosis score (**j**), serine and glycine concentration (**k**), hepatic TG and cholesterol levels (**l**) and transcriptional levels of TG synthesis/lipid storage related genes (**m**) in the liver of indicated groups, data shown as mean ± SD. **j** $n = 9$, five independent animals HFD or HFD + Ser group, respectively, $^{*}P_{H\&E} = 0.0197$; (**k–l**) $n = 9$, 6 independent animals for HFD or HFD + Ser group, respectively, $^{*}P_{serine} = 0.0377$, $^{*}P_{TG} = 0.0437$, $^{***}P_{Cholesterol} < 0.0001$; (**m**) $n = 8$, 6 independent animals for HFD or HFD + Ser group, respectively, $^{***}P_{Mogat1} = 0.0008$, $^{*}P_{Cidea} = 0.0495$, $^{**}P_{Cidec} = 0.0072$ (unpaired, two-tailed $t$-test). **n** Possible mechanisms in present study. Scale bar, 50 µm. Source data are provided in the Source Data file.

hepatocytes or HepG2 cells were treated with/without serine (200 or 400 μM) for 24 h; while serine was added once every other day with fresh medium from Day 0 to Day 6 during the SVF differentiation or at Day 6 after SVF differentiation (mature adipocytes) for 2 days, then Oil Red O staining was performed. For PHGDH inhibitor treatments, normally cultured or PA-stressed HK-2 cells were treated with 5 or 20 μM of NCT-503 (TargetMol, Boston, MA) for 24 h.

**Oil Red O staining and Nile Red Staining**. Cells or liver sections were fixed with 4% formaldehyde in PBS, and stained with 0.5% Oil Red O (Sigma) in propylene glycol[67]. Excessive staining was removed by wash with 70% ethanol, and stained cells were photographed. For Nile red staining, 0.05 mg/mL Nile red solution (Sigma) was used to visualize lipid droplets.

**Serine and glycine analysis**. Serine and glycine analysis were performed. Briefly, for tissue samples, methanol/water (1:1) was added, then homogenized, centrifuged, and supernatant was taken. For serum or medium samples, methanol was added, vortexed, centrifuged, then supernatant was taken. For urine samples, urine was centrifuged and supernatant was taken. For cell samples, 0.1 ml methanol was added to $5 \times 10^7$ cells, frozen/thawed in liquid nitrogen for three times, centrifuged and supernatant was taken. Derivatized samples were separated by a C18 analytic column on a Hitachi L-2000 HPLC equipped with a fluorescence detector and eluted with a methanol/sodium carbonate gradient, the excitation and emission wavelengths were 340 and 450 nm, respectively. Serine and glycine were quantified by standard curve method. The R2 values of linearity test were from 0.995 to 0.998, indicating good precision and accuracy.

**Quantitative Real-time PCR (qPCR) and Western Blots**. qPCR and Western blots were performed with standard procedures. Total RNA was isolated from cells using RNAiso Plus reagent (TaKaRa Biotechnology, Dalian, China) and subjected to qPCR analysis. The mRNA levels of specific genes were normalized to beta-actin. Freshly collected kidney or cultured cells were sonicated in ice-cold RIPA buffer (Beyotime), 20–80 μg protein from each sample was separated by SDS-PAGE. The proteins were transferred onto PVDF membranes for immune detection. The primers and antibodies used are provided in Supplementary Tables 8 and 10.

**Reporter Assays**. Regions from −2000 to +500 relative to transcription start site of *CIDEA* or *CIDEC* promoter was cloned into pGL3-basic vector (Promega). The HK-2 cells were transfected with pGL3-basic-CIDEA, or pGL3-basic-CIDEC, pRL-TK, and control vector or UTX plasmids for 48 h. Luciferase assays were performed and analyzed with a GloMax multimode reader (Promega)[57,68].

**Co-Immunoprecipitation (Co-IP)**. Cells were lysed with pre-lysis buffer, and cell lysates were incubated with antibodies (Supplementary Table 8) or respective IgGs with Dynabeads Protein G (Thermo Scientific, Waltham, MA) overnight at 4 °C[25]. After washing (pre-lysis buffer with additional 50 mM NaCl), the beads were boiled in loading buffer and subjected to immunoblotting.

**Chromatin immunoprecipitation (ChIP) assay**. The kidney was crosslinked using 1% formaldehyde and stopped by adding glycine, the ChIP assay was performed[69]. Chromatin was immunoprecipitated with H3K27me3 antibodies (Abcam). The purified DNA was detected by qPCR. Primer sequences are provided in Supplementary Table 11. The input samples were used as the internal control for comparison between samples.

**Microarray data analysis**. Microarray datasets deposited in NCBI Gene Expression Omnibus (GSE66494 and GSE96804) were retrieved, and analyzed via GEO

integrated analyzing tool (GEO2R) or R3.5.2 for the heatmap. Log2 fold change of serine metabolism related genes and UTX were visualized.

**Homologous modeling**. Human PHGDH structure was homologous modeled by the Swiss model (https://swissmodel.expasy.org/), with the crystal structure of *Mycobacterium tuberculosis* PHGDH (PDB 3DC2) as a reference, and illustrated by PyMol.

**16S rDNA sequencing and data analysis for gut floramicrobial**. 16S rDNA amplicon sequencing was performed on an Ion S5 XL platform (Thermo Fisher, Waltham, MA). Cluster analysis was performed using unweighted pair-group method with arithmetic means by QIIME (Version 1.7.0), phylogenetic investigation of communities by reconstruction of unobserved states was performed[70]. Sequencing data is deposited in the NCBI Sequence Read Archive (SRA) database.

**Statistics**. Data were analyzed with GraphPad Prism (version 8). Statistical analysis was performed using two-tailed Student's $t$-test for two experimental groups, and one-way ANOVA for multiple experimental groups without adjustment. Data are reported as the mean values with error bars showing the standard deviation (SD). For cell culture experiments, at least three independent experiments were performed with similar results. For animal study, the $n$ number of biological independent animals in different groups are provided in figure legends. A $p$-value of <0.05 was considered statistically significant.

## Data availability

Source data are provided with this paper. Sequencing dataset described in this work is available in NCBI SRA database under accession code SRP375452 (https://www.ncbi.nlm.nih.gov/sra/?term=SRP375452). Microarray datasets reused in this work are from NCBI Gene Expression Omnibus (GSE66494 and GSE96804). Source data are provided with this paper.

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

## Acknowledgements

Funding Statement: This work is supported by the Natural Science Foundation of China (91957114 and 32021003 to L.Z.; 31971066 to K.H.; 31500941 to H.C.), the National Key R&D Program of China (2018YFA0800700 and 2019YFA0802701 to L.Z.), the China Postdoctoral Science Foundation (2021M700050 to Y.C.), and the Natural Science Foundation of Hubei Province (2021CFA004 to K.H.; 2021CFB250 to C.L.).

## Author contributions

H.C., L.Z., and K.H. designed the study and analyzed the data. H.C., Q.W., C.L., X.Z, M.X., D.Y., Y.X., Y.Z., Y.H., C.L., performed the experiments. J.Y., H.S., and S.W. provided materials. Y.C., H.C., L.Z. and K.H. wrote the manuscript.

## Competing interests

The authors declared no competing interests.
