## [Peer Review File · Nature Communications]

REVIEWER COMMENTS

Reviewer #1 (Remarks to the Author):

Upregulation of UTX has been found in the kidneys of patients with diabetic kidney disease (DKD) or focal segmental glomerulosclerosis have been described by others. The author report that kidney specific knockout Utx is sufficient to inhibit the high-fat diet (HFD) induced lipid accumulation in the kidney and liver via upregulating circulating serine levels. UTX recruits E3 ligase RNF114 to ubiquitinate phosphoglycerate dehydrogenase (PHGDH), the rate limiting enzyme for de novo serine synthesis, resulting in serine deficiency. Oral administration of serine ameliorated HFD-induced fatty liver and renal dysfunction, suggesting a potential approach against obesity related metabolic disorders and suggesting a cross talk between the kidney and the liver. This is an overall well written manuscript that is significant as it may lead to a novel treatment option of highly prevalent high fat diet induced fatty kidney and fatty liver. While the upregulation of UTX is not novel, the mechanistic in vitro experiments are strongly supporting a role of serine and PHGDH in the protection from kidney diseases and in the cross talk between tubular cells and hepatocytes. My enthusiasm could be improved by the addition of patient characteristics as well as by the implementation of few read outs such as eGFR. A more detailed description of requests is provided below:

1- While the author claim a cross talk between the kidney and the liver is occurring, it is possible that the overall improvement of the liver phenotype in the model is mediated by improved insulin resistance in serine treated mice. This is important, even if the authors perform elegant in vitro studies of conditioned media from tubular cells and hepatocytes.

2- In the adipose tissue and liver Utx knock out mice, what was the level of serine into the circulation?

3- In the serin treated mice, what is the mechanisms for decreased glomerular hypertrophy? As stated above, it is important to check if serine administration improve glycemia and insulin resistance.

4- While human samples are being utilized, a table with patient characteristics should be included.

5- What is the eGFR of UTX KO mice? Is eGFR affecting circulating serine?

6- The author should mention in the introduction that fatty kidney disease occurs in kidney diseases of both metabolic and non metabolic origin (PMID 34341345, 31329164) and not solely in obesity related glomerulopathy.

Reviewer #2 (Remarks to the Author):

In this manuscript, Hong Chen, et al. showed a possible role of circulating serine originated from kidney tubular epithelial cells on prevention of liver steatosis in obesity-related diseases including diabetes. Knockdown of kidney Utx expression inhibited RNF114-induced ubiquitination of PHGDH, which restored PHGDH expression and serine levels in HFD-fed mice. Although the authors indicate an interesting metabolic crosstalk between kidney and liver mediated by serine, there remain some critical points that need to be addressed.

- 1. Was the kidney Utx expression increased in HFD-fed or diabetic mice or patients with obesity or diabetes?**

- 2. Regarding the mice models of kidney-specific Utx knockout, UtxKsp KO and UtxPax2KO,**
 - 1) What is the localization of Utx expression in the kidney?**
 - 2) Although the authors used HK2 cells in in vitro experiments and proximal tubular cells are most involved in lipid and glucose metabolism in the kidney, why were UtxKsp KO and UtxPax2KO mice used instead of the proximal tubular cell-specific Utx knockout mice?**
 - 3) The authors should refer to the differences between UtxKsp KO and UtxPax2KO. For example, why was the fat weight (eWAT, iWAT and BAT) increased in UtxKsp KO with normal diet and decreased in UtxKsp KO with HFD compared with WT controls, whereas the fat weight was decreased in UtxPax2KO with both normal and HFD (Suppl Table 1, 2)? Why the kidney of UtxPax2KO with HFD was smaller compared with controls?**

- 3. What is the detailed mechanism by which HFD-induced lipid accumulation in the kidney was ameliorated in the Utx KO mice, mediated by serine? Was the alteration of glucose or fatty acid metabolism observed in the kidney of Utx KO mice?**

- 4. In the discussion section, the authors referred a previously reported role of liver PHGDH on the pathogenesis of NAFLD, then why the liver-specific Utx KO did not cause NAFLD? Liver Utx is not involved in PHGDH regulation?**

- 5. Decreased fat weight may be one of the primary causes of weight loss in Utx KO mice, and what is the mechanism? Is it due to an increase in circulating serine as in the liver?**

- 6. As for glycine, does it affect NAFLD or weight change?**

Reviewer #3 (Remarks to the Author):

The manuscript by Chen et al. describes a pathway that includes UTX and the serine biosynthetic enzyme PHGDH that is an important modulator of serine levels, which in turn alters lipid accumulation. The authors show that kidney specific knockout of the H3K27 demethylase UTX attenuates the effects of a high fat diet, not only within the kidney, but also in cells of the liver and adipose tissue. This very interesting observations suggests both cell autonomous and cell non-autonomous effects caused by loss of UTX in the kidney, which is likely mediated through altered circulating serine levels. While the work provides new and interesting data, from the perspective of UTX function, it is unclear exactly what exactly is being proposed. Does the proposed UTX-RNF114-PHGDH complex regulate transcription? Does this function require the demethylase activity of UTX? Does UTX recruitment to its target genes change in response to a high fat diet (in the kidney or elsewhere)? The data, as they stand, don't really go far enough to propose a coherent model.

Other more minor concerns:

- Figure 5a, b, data from published microarray data are presented. However, it is how these analyses were done or what the red/blue scale indicates. The materials and methods indicate that it is log₂FC, which raises the question of why controls are increased – shouldn't they be unchanged. Are the rows different individuals? If so, why does there appear to be no variation for CDK in part a?
- Also related to 5a, b. It would be very useful to know what the levels of UTX were in these same individuals.
- The data shown in figure 2 e, f, are overinterpreted. While levels of Cidea and Cidec mRNA are reduced in UtxKsp mice fed a normal diet, this is not sufficient to conclude that these genes are directly regulated by UTX (p7). That conclusion would require additional experiments such as ChIP.
- The rationale for using palmitic acid is not stated (Figure 6). The acronym PA also needs to be defined at the first use (p9), not just in the materials and methods.
- In Figure 6h-i, PHGDH KO alone should also be examined, not just the double mutant.
- Also in Figure 6, how specific is the NCT-503 inhibitor? It may be useful to add a control of PHGDH KO cells treated with NCT-503.
- Related to Figure 7, what is the level of overexpression of UTX (RNF114 and PHGDH) when transiently transfected? It is a major concern that overexpressed proteins may not behave the same way as endogenous proteins. I have a similar concern regarding the ubiquitination assays, all of which use overexpressed proteins.
- Why doesn't the model shown in Figure 9 include the proposed physical interaction between UTX/RNF114 and PHGDH?
- The manuscript uses a lot of acronyms. Please consider reducing the number to broaden the appeal of the manuscript to those beyond who are not already experts in kidney function.

Reviewer #4 (Remarks to the Author):

This interesting but poorly written manuscript by Chen et al characterizes the role of kidney Utx in the development of diet-induced obesity. They found that deletion of Utx in the kidney results in resistance to high fat diet-induced metabolic dysfunctions. The Utx KO mice showed lower body weight, improved performance in glucose and insulin tolerance tests, and reduced hepatic steatosis. These mice also showed lower expression of PHGDH and serine in the kidney. Mechanistically, the authors showed that Utx modulates the stability of PHGDH through ubiquitination. While the phenotypes described and the function of Utx in the regulation of protein stability are novel, critical links are missing regarding the mechanism proposed, and I am not convinced about the importance of direct kidney to liver crosstalk that does not involve other organs, dramatically lowering enthusiasm for the study.

1. In Figure 1, the mice are leaner with no change in food intake. Are the mice in general healthy? Is the activity level comparable between WT and KO?
2. figures 1 and 2 showed dysfunctions in the kidney. A panel of function analysis of the kidney should also be included, such as urine albumin to creatinine ratio and BUN level, to show the

injury in the kidney and the possible effect on the overall health of the KO mice.

3. Do the changes in kidney function and lipid accumulation of UTX KO mice affect kidney gluconeogenesis and overall reabsorption and excretion of glucose?

4. Except for the genes shown in TG and cholesterol, are there changes in SREBP, FXR, PPAR α , and LDLR mRNA expression and protein levels?

5. In Figure 4, the deletion of Utx also greatly affects the adipose tissue, which potentially affects hepatic lipid metabolism. It is therefore very hard to conclude that the target tissue of kidney serine is the liver based on only the condition medium assay. The title and conclusion regarding the kidney-liver crosstalk are therefore not accurate. This speaks to the issues regarding mechanism in vivo.

6. In Figure 5, the authors showed that PHGDH is decreased in obesity and CKD patients. How is Utx changed in these patients?

7. The authors proposed that Utx affects PHGDH protein stability. However, no change is observed in mice fed with normal chow diet in Figure 5. How is PHGDH differently regulated by Utx under these two conditions? In other words, what is the trigger in HFD that causes the changes in either PHGDH or Utx?

8. Related to question 8, Utx and PHGDH are widely expressed in liver and adipose tissues. However, no phenotype is observed in liver and adipose tissue Utx KO mice. What causes the tissue specificity in the kidney?

9. In Figure 6, what causes the difference in 6c and 6f regarding the PHGDH level in the control group? In 6c, there is no change in PHGDH level with PA treatment. However, PHGDH basal level is significantly higher in the ctrl group and decreased with PA treatment in 6f. Similarly, what causes the difference in 6h and 6n regarding the fold change between ctrl and utx ko group with PA treatment?

10. In both Figure 8 and 9, where PHGDH is overexpressed or serine supplemented, how is the cellular NAD⁺/NADH level changed?

The English grammar is weak throughout the manuscript and it needs extensive editing.

REVIEWER COMMENTS

Reviewer #1

Upregulation of UTX has been found in the kidneys of patients with diabetic kidney disease (DKD) or focal segmental glomerulosclerosis have been described by others. The author report that kidney specific knockout Utx is sufficient to inhibit the high-fat diet

(HFD) induced lipid accumulation in the kidney and liver via upregulating circulating serine levels. UTX recruits E3 ligase RNF114 to ubiquitinate phosphoglycerate dehydrogenase (PHGDH), the rate limiting enzyme for *de novo* serine synthesis, resulting in serine deficiency. Oral administration of serine ameliorated HFD-induced fatty liver and renal dysfunction, suggesting a potential approach against obesity related metabolic disorders and suggesting a cross talk between the kidney and the liver. This is an overall well written manuscript that is significant as it may lead to a novel treatment option of highly prevalent high fat diet induced fatty kidney and fatty liver. While the upregulation of UTX is not novel, the mechanistic *in vitro* experiments are strongly supporting a role of serine and PHGDH in the protection from kidney diseases and in the cross talk between tubular cells and hepatocytes. My enthusiasm could be improved by the addition of patient characteristics as well as by the implementation of few read outs such as eGFR. A more detailed description of requests is provided below.

Response: We appreciate the reviewer for the encouraging comments that “This is an overall well written manuscript that is significant as it may lead to a novel treatment option of highly prevalent high fat diet induced fatty kidney and fatty liver.” As the reviewer suggested, in the revised manuscript, we provided additional available patient characteristics, such as eGFR, TG, TC, albumin, globin and BUN, in revised Supplementary Tables 1, 4-5. We also performed new experiments and revised certain texts to address the reviewer’s comments accordingly.

1. While the authors claim a cross talk between the kidney and the liver is occurring, it is possible that the overall improvement of the liver phenotype in the model is mediated by improved insulin resistance in serine treated mice. This is important, even if the authors perform elegant in vitro studies of conditioned media from tubular cells and hepatocytes.

Response: We thank the reviewer for this important and insightful suggestion. In the revised manuscript, we performed the GTT/ITT tests on HFD-fed mice, with or without serine treatment; moreover, we detected the insulin signaling such as p-AKT and p-GSK 3 β in the liver (revised Supplementary Fig. 19). All these data indicated that the serine treatment, at the dosage and duration of our experimental setting, showed no obvious improvement on insulin resistance in HFD-fed mice.

2. In the adipose tissue and liver *Utx* knock out mice, what was the level of serine into the circulation?

Response: As the reviewer suggested, in the revised manuscript, we measured the serum serine levels of HFD-fed liver knockout *Utx* (*Utx^{Alb}* KO) mice and their respective control mice. No significantly altered serum serine level was observed in *Utx^{Alb}* KO mice vs. the control mice (revised Supplementary Fig. 9g). Due to the lack of obvious phenotype in HFD-fed adipose tissue specific *Utx* knockout (*Utx^{Adi}* KO) mice, this line was not kept in the lab anymore, and no blood sample was available for serine measurement. Instead, we

stained PHGDH in the kidney, eWAT, iWAT and BAT tissues that we previously collected from the NC- or HFD-fed *Utx^{Adi}* KO males. The staining shows similar PHGDH level in these tissues between the *Utx^{Adi}* KO and the control mice under NC- and HFD-feeding conditions (revised Supplementary Fig. 9h). Taken together, these data indicate that liver or adipose tissue specific knockout *Utx* may not affect serum serine level.

3. In the serine treated mice, what is the mechanisms for decreased glomerular hypertrophy? As stated above, it is important to check if serine administration improve glycemia and insulin resistance.

Response: Previous studies have indicated that lipotoxicity and systemic insulin resistance as major mechanisms involve in HFD-induced glomerular hypertrophy (Megalin-Mediated Tubuloglomerular Alterations in High-Fat Diet-Induced Kidney Disease. *J Am Soc Nephrol.* 2016 27(7):1996-2008; Kidney glycosphingolipids are elevated early in diabetic nephropathy and mediate hypertrophy of mesangial cells, *Am J Physiol Renal Physiol.* 2015, 309: F204-215). In the revised manuscript, as the reviewer suggested, we investigated the possible mechanisms for decreased glomerular hypertrophy. While the results suggested no significant effects of serine treatment on improving systemic glycemia and insulin resistance (revised Supplementary Fig. 19a-b), our studies indicated that reduced lipotoxicity may benefit glomerular hypertrophy. Because when we knocked out serine transporter *ASCT1* and/or *ASCT2* in serine treated HK-2 cells,

beneficial effects of serine on lipotoxicity such as down-regulated TG content, transcriptions of TG synthesis and lipid storage genes, were abolished (revised Fig. 9 c-f).

4. While human samples are being utilized, a table with patient characteristics should be included.

Response: We thank the reviewer for this kind reminder. In the previous manuscript, we provided some basic patients characteristics in the previous Supplementary Tables. As the reviewer suggested, we included additional available patient characteristics, such as eGFR, TG, TC, albumin, globin, BUN in the revised tables. Revised supplementary Tables 1 and 4 provide individual patients' information, renal samples from these patients were used for immunohistochemical studies. Revised supplementary Table 5 provides a summary for different patient groups whose serum were used for the serine/glycine measurements, due to the relatively large number (98 cases) of patients included for these measurements, individual patient's information was provided in the raw data.

5. What is the eGFR of UTX KO mice? Is eGFR affecting circulating serine?

Response: In the revised manuscript, we provided the eGFR of mice calculated by the creatinine clearance with the formula $Ccr = \frac{CREA(urine) * Volume}{CREA (serum) / 1440}$ (24 hours) as described in the literature (Four-hour creatinine clearance is better than plasma creatinine for monitoring renal function in critically ill patients, *Crit Care*. 2012

16(3): R107; Histone demethylase UTX is a therapeutic target for diabetic kidney disease, *J Physiol.* 2019 597(6): 1643–1660). No significant difference in the eGFR was found between the WT and *Utx^{Ksp}* KO mice, under either NC- or HFD-fed conditions (revised Supplementary Fig.1n).

Whether eGFR affects circulating serine level is an interesting question. In the present study, while there was no significant change in the eGFR level between the WT and *Utx^{Ksp}* KO mice under NC- and HFD-fed conditions (revised Supplementary Fig. 1n), HFD-fed *Utx* KO mice showed increased circulating serine level (Fig. 5m), indicating no correlation between eGFR and circulating serine level at our experimental setting. We further investigated the eGFR and circulating serine level in collected clinical samples. Interestingly, a mild positive correlation between eGFR and circulating serine level was suggested in these clinical samples (revised Fig. 5h). Therefore, it will be interesting to explore as a future direction in a larger cohort among different normal and renal disease groups. This interesting point was also discussed in the revised Discussion (page 19).

6. The author should mention in the introduction that fatty kidney disease occurs in kidney diseases of both metabolic and non metabolic origin (PMID 34341345, 31329164) and not solely in obesity related glomerulopathy.

Response: Thanks for this kind advice. We have revised this sentence and cited these references in the revised Introduction (page 4).

Reviewer #2

In this manuscript, Hong Chen, et al. showed a possible role of circulating serine originated from kidney tubular epithelial cells on prevention of liver steatosis in obesity-related diseases including diabetes. Knockdown of kidney Utx expression inhibited RNF114-induced ubiquitination of PHGDH, which restored PHGDH expression and serine levels in HFD-fed mice. Although the authors indicate an interesting metabolic crosstalk between kidney and liver mediated by serine, there remain some critical points that need to be addressed.

Response: We thank the reviewer for the encouraging comments that “the authors indicate an interesting metabolic crosstalk between kidney and liver mediated by serine.” We have tried our best to address the reviewer’s comments by performing new experiments.

1. Was the kidney Utx expression increased in HFD-fed or diabetic mice or patients with obesity or diabetes?

Response: Previous studies by others and us have reported increased renal UTX expression in patients with diabetic kidney disease (DKD) or focal segmental glomerulosclerosis, as well as in the kidneys of diabetic mouse models including Akita mice, db/db mice, OVE26, and STZ-treated mice (Shifts in podocyte histone H3K27me3 regulate mouse and human glomerular disease, *J Clin Invest.* 2018 128(1): 483-499; Histone demethylase UTX is a

therapeutic target for diabetic kidney disease, *J Physiol.* 2019 597(6): 1643-1660; Epigenetic changes in renal genes dysregulated in mouse and rat models of type 1 diabetes, *Laboratory Investigation.* 2013, 93(5): 543–552).

As the reviewer suggested, in the revised manuscript, we further performed UTX staining in the kidneys of HFD-fed mice (cryosections), as well as in the kidney samples of the obese patients (paraffin embedded sections). Both results suggested increased UTX expression in tubular cells and glomeruli in the kidneys of HFD-fed mice and obese patients (revised Fig. 1a-b).

2. Regarding the mice models of kidney-specific Utx knockout, UtxKsp KO and UtxPax2 KO

1) What is the localization of Utx expression in the kidney?

Response: Based on previous studies by others and us, UTX locates in the podocytes, mesangial and tubular cells (Shifts in podocyte histone H3K27me3 regulate mouse and human glomerular disease, *J Clin Invest.* 2018 128(1): 483-499; Histone demethylase UTX is a therapeutic target for diabetic kidney disease, *J Physiol.* 2019 597(6): 1643-60). As the reviewer suggested, we performed additional co-immunostaining studies to show the localization of UTX in the kidney under NC and HFD stress. These new results demonstrated weak UTX expression in the glomeruli, proximal tubules, and distal

tubules/collecting tubules, which were significantly up-regulated under HFD stress (revised Fig. 1b).

2) Although the authors used HK2 cells in *in vitro* experiments and proximal tubular cells are most involved in lipid and glucose metabolism in the kidney, why were Utx^{Ksp} KO and Utx^{Pax2} KO mice used instead of the proximal tubular cell-specific *Utx* knockout mice?

Response: Since others and our results have shown increased UTX staining in multiple cell types of the kidney (revised Fig. 1a-b; Shifts in podocyte histone H3K27me3 regulate mouse and human glomerular disease, *J Clin Invest.* 2018 128(1): 483-499; Histone demethylase UTX is a therapeutic target for diabetic kidney disease, *J Physiol.* 2019 597(6): 1643-60), we initially generated whole kidney specific *Utx* knockout mice using Pax2-cre. As the studies progressed, we realized that the renal tubular cells may play an important role in the resistance to HFD stress, so we generated the renal tubular specific *Utx* knockout mouse using Ksp-cre, which has previously been reported to also express in renal proximal tubular cells (revised Fig. 1f; RGMb protects against acute kidney injury by inhibiting tubular cell necroptosis via an MLKL-dependent mechanism, *Proc Natl Acad Sci U S A.* 2018 115(7):E1475-84; Tubule-specific ablation of endogenous β -catenin aggravates acute kidney injury in mice, *Kidney Int.* 2012 82(5):537-47; Rheb1 protects against cisplatin-induced tubular cell death and acute kidney injury via maintaining

mitochondrial homeostasis, *Cell Death Dis.* 2020 11(5):364). Our staining on renal cryosections also demonstrated Ksp-cre successful knockout Utx in the proximal tubules, as well as distal tubules (revised Fig. 1f). We agree with the reviewer that proximal tubular cells are mainly involved in lipid and glucose metabolism in the kidney, therefore it will be interesting as a future study to use a renal proximal tubular specific Cre, such as SGLT2-cre, to generate proximal tubular specific *Utx* knockout mice, and study its role in lipid metabolism. We also addressed this issue in the revised Discussion (page 17).

3) The authors should refer to the differences between Utx^{Ksp} KO and Utx^{Pax2} KO. For example, why was the fat weight (eWAT, iWAT and BAT) increased in Utx^{Ksp} KO with normal diet and decreased in Utx^{Ksp} KO with HFD compared with WT controls, whereas the fat weight was decreased in Utx^{Pax2} KO with both normal and HFD (Suppl Table 1, 2)? Why the kidney of Utx^{Pax2} KO with HFD was smaller compared with controls?

Response: Thanks for this important suggestion. As the reviewer suggested, in the revised manuscript, we further compared the differences between the Utx^{Ksp} KO and Utx^{Pax2} KO mice. In the kidney, Ksp-cre is reported to express in renal proximal and distal tubulars, collecting ducts, loops of Henle (Epithelial-specific cre/lox recombination in the developing kidney and genitourinary tract, *J Am Soc Nephrol.* 2002 13:1837-46); whereas the Pax2-cre expression is detected in renal proximal and distal tubulars, glomerular, mesenchyme, loops of Henle, collecting ducts, nephric duct of the developing and adult

kidney (Kidney-specific gene targeting, *J Am Soc Nephrol.* 2004, 15:2237-39; <https://www.jax.org/strain/012237>; Generation of Pax2-Cre Mice by Modification of a Pax2 Bacterial Artificial Chromosome, *Ggenesis.* 2004, 38:195–199). These differences may produce some variability in biological functions caused by knockout of Utx in varied renal cell types mediated by different Cre strains. Similar phenomena have been previously observed in knockout of other genes in the same tissue mediated by different Cre strains. For example, in adipose tissues for *aP2*-cre and adiponectin-cre. *AMPK^{ap2}* KO causes decreased kidney weight, while *AMPK^{Adiponectin}* KO shows no effect on kidney weight in the mice (AMPK phosphorylates desnutrin/ATGL and hormone-sensitive lipase to regulate lipolysis and fatty acid oxidation within adipose tissue, *Mol Cell Biol.* 2016 36(14):1961-76). Moreover, *Ckmm*-cre or *Myhca*-cre driven *Tfam* heart specific knockout mice also exert some different functions. The *Tfam^{Ckmm}* KO mice show normal respiratory chain function in the heart at birth and developed mitochondrial cardiomyopathy postnatally, while the *Tfam^{Myhca}* KO mice show onset of cardiomyopathy during embryogenesis (Genetic modification of survival in tissue-specific knockout mice with mitochondrial cardiomyopathy, *Proc Natl Acad Sci USA*, 2000; 97(7): 3467–3472). We also addressed this issue in the revised Discussion (page 18).

3. What is the detailed mechanism by which HFD-induced lipid accumulation in the kidney was ameliorated in the Utx KO mice, mediated by serine? Was the alteration of glucose or fatty acid metabolism observed in the kidney of Utx KO mice?

Response: Thanks for this important and insightful suggestion. As the reviewer required, in the revised manuscript, we first investigated whether the observed ameliorated lipid accumulation in renal cells is serine-dependent. In PA-treated HK-2 cells, serine treatment reduced lipid accumulation and transcription levels of genes involved in TG synthesis/storage; however, when we knocked out serine transporter, ASCT1 and/or ASCT2, serine treatment no longer showed beneficial effects (revised Fig. 9c-f). It has been reported that serine increases intra-cellular NAD⁺ level that leads to SIRT1 activation, which regulates lipid metabolism through affecting the acetylation of PGC1 α and mitochondrial function (Activation of SIRT1 by L-serine increases fatty acid oxidation and reverses insulin resistance in C2C12 myotubes, *Cell Biol Toxicol.* 2019 35(5):457-70). Thus, we measured NAD⁺/NADH in culture HK-2 cells, and found serine administration significantly increased NAD⁺/NADH level (revised Supplementary Fig. 15).

As the reviewer suggested, we further examined the relative gene levels of lipid synthesis, lipid transport, beta oxidation and glucose metabolism pathways in the kidneys of *Utx*^{Ksp} KO mice. Under HFD-feeding, compared with the WT mice, *Utx*^{Ksp} KO mice showed significantly decreased *Fasn*, *Glut4*, *G6pc*, and *Pc* mRNA levels, whereas the other genes levels were not changed (revised Supplementary Fig. 4 b-f).

4. In the discussion section, the authors referred a previously reported role of liver PHGDH on the pathogenesis of NAFLD, then why the liver-specific *Utx* KO did not cause NAFLD? Liver *Utx* is not involved in PHGDH regulation?

Response: We thank the reviewer for this kind reminder. According to the reviewer's suggestion, we further detected the PHGDH protein level in the liver of *Utx^{Alb}* KO mice. The WB results showed that liver knockout *Utx* did not affect PHGDH level (revised Supplementary Fig. 9). According the Human Protein Atlas websites (<https://www.proteinatlas.org/ENSG00000124226-RNF114/tissue>) and a new Western blots in mouse tissues (data shown below), and found the protein level of RNF114 in the kidney was higher than that in the liver and the adipose tissue. These data may explain the absence of obvious effect of liver-specific *Utx* KO on PHGDH. We also described this issue in the revised Discussion (page 18).

5. Decreased fat weight may be one of the primary causes of weight loss in Utx KO mice, and what is the mechanism? Is it due to an increase in circulating serine as in the liver?

Response: Thanks for this insightful comment. As the reviewer's suggested, we further performed conditional culture medium exchange experiment between HK-2 cells and primary mouse SVF cells. Under PA-stress, compared to cells treated with medium from respective control cells, reduced lipid accumulation was found in the SVF cells treated with medium from UTX KO cells (data shown below). To further explore whether reducing lipid accumulation of adipose cells by medium from UTX KO HK-2 cells is through serine, as we observed in conditional culture medium exchange between HK-2 cells and primary mouse hepatocytes, serine was administrated to the medium of primary mouse SVF cells for differentiation or mature adipocytes after differentiation. The results suggested that serine did not affect adipocyte differentiation or lipid accumulation in mature adipocytes (revised Supplementary Fig. 18). Together, these results suggested that renal knockout of Utx does release unknown factor(s) that may affect adipose tissue, however, serine is not the one. We also discussed this issue in the revised Discussion (page 20-21).

6. As for glycine, does it affect NAFLD or weight change?

Response: Thanks for this great suggestion. In the revised manuscript, as the reviewer suggested, we administrated glycine to PA-treated hepatocytes, and found that glycine downregulated TG accumulation (revised Supplemental Fig. 17). Consistently, a recent study has suggested that glycine treatment ameliorates NAFLD (Glycine-based treatment ameliorates NAFLD by modulating fatty acid oxidation, glutathione synthesis, and the gut microbiome, *Sci Trans Med.* 2020 Dec 2;12(572): eaaz2841).

Reviewer #3

The manuscript by Chen et al. describes a pathway that includes UTX and the serine biosynthetic enzyme PHGDH that is an important modulator of serine levels, which in turn alters lipid accumulation. The authors show that kidney specific knockout of the H3K27 demethylase UTX attenuates the effects of a high fat diet, not only within the kidney, but also in cells of the liver and adipose tissue. This very interesting observations suggests both cell autonomous and cell non-autonomous effects caused by loss of UTX in the kidney, which is likely mediated through altered circulating serine levels. While the work provides new and interesting data, from the perspective of UTX function, it is unclear exactly what exactly is being proposed.

Response: We thank the reviewer for the encouraging comments that “This very interesting observations suggests both cell autonomous and cell non-autonomous effects caused by

loss of UTX in the kidney, which is likely mediated through altered circulating serine levels.” We have tried our best to improve the manuscript by performing new experiments and revising the text to address the reviewer’s comment.

1. Does the proposed UTX-RNF114-PHGDH complex regulate transcription? Does this function require the demethylase activity of UTX? Does UTX recruitment to its target genes change in response to a high fat diet (in the kidney or elsewhere)? The data, as they stand, don’t really go far enough to propose a coherent model.

Response: We thank the reviewer for this important suggestion. To the best of our knowledge, current literatures indicated that UTX may directly regulate the transcription, while PHGDH and RNF114 showed no evidence of direct transcriptional regulatory activity. Consistently, in PHGDH or RNF114 KO cells, no change in UTX level was observed (revised Supplementary Fig. 14d).

UTX has been reported to promote gene transcription by removing H3K27me3 from the promoters. We have tried multiple UTX antibodies for ChIP assay without success, therefore, we performed a ChIP assay to examine the enrichment of H3K27me3 on the promoters of *Cidea* and *Cidec* genes in the kidneys of *Utx^{pax2}* KO mice. The results showed H3K27me3 enrichment on the promoters of *Cidea* and *Cidec* genes (revised Supplementary Fig. 3d), which suggested that UTX may be recruited to these target genes. To investigate whether the demethylase activity of UTX is required for the regulation of

CIDEA and *CIDEC*, plasmids expressing UTX WT, or a catalytic-domain mutated UTX, was respectively transfected into UTX KO HK-2 cells. Overexpression of UTX WT but not mutated UTX increased *CIDEA* and *CIDEC* levels, indicating that the regulation of UTX in lipid accumulation depends on its demethylase activity (revised Supplementary Fig. 10e). Furthermore, an *in vitro* luciferase reporter assay confirmed the up-regulation of UTX on *CIDEA* and *CIDEC* (revised Supplementary Fig. 10f).

Other more minor concerns:

1. Figure 5a, b, data from published microarray data are presented. However, it is how these analyses were done or what the red/blue scale indicates. The materials and methods indicate that it is log₂FC, which raises the question of why controls are increased – shouldn't they be unchanged. Are the rows different individuals? If so, why does there appear to be no variation for CDK in part a?

Response: We thank the reviewer for this kind reminder. We retrieved these microarray data from the NCBI Gene Expression Omnibus (GSE66494 and GSE96804), and analyzed *via* GEO integrated analyzing tool (GEO2R) or R3.5.2 for the heatmap. The red scale indicates the positive log₂FC value for up-regulated genes, while blue scale indicates the negative values for down-regulated genes. The rows indicate different individuals. Because the values for each sample in the CKD group were very close, the small variation

was thus not highly distinguishable in the figure. We have re-checked the retrieved raw data from the GEO database, which is attached below.

		PHGDH	PSAT1	PSPH	SHMT1	SHMT2	UTX
GSM1623352	Validation_CKD_01	1.732146	5.184902	32.63498	12.76542	0.800636	4.717044
GSM1623353	Validation_CKD_02	3.704319	9.322088	167.2941	15.7958	1.643717	13.1228
GSM1623354	Validation_CKD_03	4.603693	27.79252	98.41791	71.59068	6.695844	11.39641
GSM1623355	Validation_CKD_04	6.197613	46.43919	81.54688	23.19789	3.386615	17.81443
GSM1623356	Validation_CKD_05	1.60165	8.029815	114.8922	8.062011	1.311387	13.54987
GSM1623357	Validation_Control_01	37.99827	53.81943	65.81268	34.83949	13.46719	4.452924
GSM1623358	Validation_Control_02	35.76489	52.18163	66.17503	35.16444	12.39839	4.373616
GSM1623359	Validation_Control_03	35.47171	52.70734	67.08793	35.29462	12.76658	4.354568

2. Also related to 5a, b. It would be very useful to know what the levels of UTX were in these same individuals.

Response: Thanks for this great suggestion. As the reviewer suggested, we have included the transcriptional level of *UTX* in the same microarray data (revised Fig. 5a-b).

3. The data shown in figure 2 e, f, are overinterpreted. While levels of *Cidea* and *Cidec* mRNA are reduced in *Utx*^{Ksp} mice fed a normal diet, this is not sufficient to conclude that these genes are directly regulated by UTX (p7). That conclusion would require additional experiments such as ChIP.

Response: We thank the reviewer for this great suggestion. The mRNA levels of *Cidea* and *Cidec* were reduced in both *Utx*^{Ksp} and *Utx*^{Pax2} KO mice fed a normal diet. With no kidney samples of *Utx*^{Ksp} mice currently available, in the revised manuscript, we performed ChIP assay to examine the enrichment of H3K27me3 on *Cidea* and *Cidec* promoters in the

kidneys of *Utx*^{Pax2} KO mice, and demonstrated increased H3K27me3 enrichment on the promoters of *Cidea* and *Cidec* genes (revised Supplementary Fig. 3d).

4. The rationale for using palmitic acid is not stated (Figure 6). The acronym PA also needs to be defined at the first use (p9), not just in the materials and methods.

Response: Thanks for this reminder. As the reviewer suggested, the rationale for using palmitic acid was described in the revised manuscript, and its acronym was also defined at the first use (page 11).

5. In Figure 6h-i, PHGDH KO alone should also be examined, not just the double mutant.

Response: We thank the reviewer for this kind reminder. In the previous manuscript, PHGDH KO related results were provided in Fig. 8d-f. As the reviewer suggested, we have moved the PHGDH KO related results to Fig. 6h-k in the revised manuscript.

6. Also in Figure 6, how specific is the NCT-503 inhibitor? It may be useful to add a control of PHGDH KO cells treated with NCT-503.

Response: Many thanks for this great suggestion. In the revised manuscript, as the reviewer suggested, we included an NCT-503 treated PHGDH KO group to examine the specificity of the inhibitor. The results indicated that NCT-503 treatment could not further down-regulate serine levels in PHGDH KO cells (revised Supplementary Fig. 12).

7. Related to Figure 7, what is the level of overexpression of UTX (RNF114 and PHGDH) when transiently transfected? It is a major concern that overexpressed proteins may not behave the same way as endogenous proteins. I have a similar concern regarding the ubiquitination assays, all of which use overexpressed proteins.

Response: We appreciate the reviewer for this important suggestion. As the reviewer suggested, in the revised manuscript, we further detected the level of UTX, PHGDH and RNF114 under transiently transfection, and dramatically increased UTX, PHGDH and RNF114 were found (revised Supplementary Fig. 14a-c). We agree with the reviewer that overexpressed proteins may not behave the same way as endogenous proteins. Therefore, in the revised manuscript, we performed the endogenous UTX/RNF114/PHGDH binding assay and ubiquitination assay with or without PA treatment (revised Figure 8e and 8i), the same conclusion was drawn as we obtained in the stable cell lines.

8. Why doesn't the model shown in Figure 9 include the proposed physical interaction between UTX/RNF114 and PHGDH?

Response: We thank the reviewer for this great suggestion. We have re-drawn this mechanistic model in the revised manuscript (revised Fig. 10n).

9. The manuscript uses a lot of acronyms. Please consider reducing the number to broaden the appeal of the manuscript to those beyond who are not already experts in kidney function.

Response: As the reviewer suggested, we have reduced the number of acronyms and explained remaining necessary acronyms more clearly.

Reviewer #4

This interesting but poorly written manuscript by Chen et al characterizes the role of kidney Utx in the development of diet-induced obesity. They found that deletion of Utx in the kidney results in resistance to high fat diet-induced metabolic dysfunctions. The Utx KO mice showed lower body weight, improved performance in glucose and insulin tolerance tests, and reduced hepatic steatosis. These mice also showed lower expression of PHGDH and serine in the kidney. Mechanistically, the authors showed that Utx modulates the stability of PHGDH through ubiquitination. While the phenotypes described and the function of Utx in the regulation of protein stability are novel, critical links are missing regarding the mechanism proposed, and I am not convinced about the importance of direct kidney to liver crosstalk that does not involve other organs, dramatically lowering enthusiasm for the study.

Response: We thank the reviewer for his/her comments that the manuscript is interesting and “the phenotypes described and the function of Utx in the regulation of protein stability

are novel”. We have tried our best to improve the manuscript according to the reviewer’s comments and suggestions by performing new experiments and revising the text.

1. In Figure 1, the mice are leaner with no change in food intake. Are the mice in general healthy? Is the activity level comparable between WT and KO?

Response: The *Utx* KO mice were generally healthy. All animals were born normally, and displayed no obvious behavioral abnormalities. During the experimental period we examined, we detected the activity of WT and *Utx*^{Ksp/Pax2} KO mice under NC- and HFD-fed conditions by using CLAMS (Columbus Instruments). Compared with WT mice, *Utx*^{Ksp/Pax2} KO mice showed more movements during the light time (data shown below).

2. figures 1 and 2 showed dysfunctions in the kidney. A panel of function analysis of the kidney should also be included, such as urine albumin to creatinine ratio and BUN level, to show the injury in the kidney and the possible effect on the overall health of the KO mice.

Response: As the reviewer suggested, we have measured the level of uACR and uBUN of the NC- or HFD-fed *Utx*^{Ksp} KO mice. Compared with NC-fed WT mice, HFD-feeding induced an increase in the level of uBUN and uACR; whereas knockout of *Utx* resulted in a trend of decreased uACR (p = 0.08) under HFD stress (revised Supplementary Fig. 11-n).

3. Do the changes in kidney function and lipid accumulation of UTX KO mice affect kidney gluconeogenesis and overall reabsorption and excretion of glucose?

Response: As the reviewer suggested, in the revised manuscript, we have detected the transcriptional levels of genes involved in the regulation of gluconeogenesis as well as those involved in the reabsorption and excretion of glucose, including *G6pc*, *Fbp*, *Pepck*, *Pc*, *Gapdh*, *Gpi*, *Pgam*, *Aloda*, *Sgt1*, *Sgt2* and *Glut4*. These new data suggested that kidney knockout *Utx* decreased *G6pc*, *Pc* and *Glut4* mRNA levels, but did not affect other genes detected (revised Supplementary Fig. 4e-f).

4. Except for the genes shown in TG and cholesterol, are there changes in SREBP, FXR, PPAR α , and LDLR mRNA expression and protein levels?

Response: Thanks for this great suggestion. In the revised manuscript, we have detected the mRNA and/or protein levels of SREBP, FXR, PPAR α and LDLR. This new result suggested similar levels of SREBP, FXR, PPAR α and LDLR in the kidneys of WT and *Utx*^{Ksp} KO mice (revised Supplementary Fig. 4a-d).

5. In Figure 4, the deletion of *Utx* also greatly affects the adipose tissue, which potentially affects hepatic lipid metabolism. It is therefore very hard to conclude that the target tissue of kidney serine is the liver based on only the condition medium assay. The title and conclusion regarding the kidney-liver crosstalk are therefore not accurate. This speaks to the issues regarding mechanism *in vivo*.

Response: Thanks for this great suggestion. As the reviewer suggested, in the revised manuscript, we have explored the effect of serine treatment on adipose tissue. *In vitro*, serine treatment did not affect primary mouse SVF cells differentiation, and lipid accumulation in primary mouse mature adipocytes (revised Supplementary Fig. 18). Moreover, *in vivo* serine treatment showed no obvious effect on adipose hypertrophy and adipose tissue weights under HFD stress (revised Supplementary Fig. 20 and Supplementary Table 6). Since deletion of renal *Utx* does affect the adipose tissue, we further performed conditional culture medium exchange experiment between HK-2 cells and primary mouse SVF cells. Under PA-stress, compared to cells treated with medium from respective control cells, reduced lipid accumulation was found in the SVF cells treated with medium from UTX KO cells (data shown below). Together, these results suggested that renal knockout of *Utx* does release unknown factor(s) that may affect adipose tissue, however, serine is not the one. The effect of renal knockout *Utx* on adipose

tissue warrants further investigation in the future. We also discussed this issue in the revised Discussion (page 20-21).

6. In Figure 5, the authors showed that PHGDH is decreased in obesity and CKD patients. How is Utx changed in these patients?

Response: In the previous Figure 5, we showed that transcriptional level of *PHGDH* was decreased in CKD and DKD patients in microarray data, while protein level of PHGDH was decreased in the kidneys of DKD patients and HFD-fed mice. As the reviewer suggested, we have included the transcriptional level of *UTX* in the same microarray data (revised Fig. 5a-b). Since previous studies by others and us have reported increased renal *UTX* expression in patients with diabetic kidney disease (DKD) or focal segmental glomerulosclerosis (Shifts in podocyte histone H3K27me3 regulate mouse and human glomerular disease, *J Clin Invest.* 2018 128(1): 483-499; Histone demethylase UTX is a therapeutic target for diabetic kidney disease, *J Physiol.* 2019 597(6): 1643-1660), in the revised manuscript, we further performed UTX staining in the kidney samples of the obese patients (paraffin embedded sections). The new piece of data showed increased protein level of UTX in the tubular cells and glomeruli in the kidney of obese patients (revised Fig.

1a). Moreover, we also stained PHGDH in the kidney samples of the obese patients (paraffin embedded sections) and found its significant down-regulation (revised Fig. 5c).

7. The authors proposed that Utx affects PHGDH protein stability. However, no change is observed in mice fed with normal chow diet in Figure 5. How is PHGDH differently regulated by Utx under these two conditions? In other words, what is the trigger in HFD that causes the changes in either PHGDH or Utx?

Response: Thanks for pointing out this great issue. Under HFD condition, UTX expression was significantly increased in the kidney. Our data (Supplementary Fig. S13b) and a single-cell sequencing study suggest upregulated RNF114 under obesity or diabetic conditions in the kidney (The single-cell transcriptomic landscape of early human diabetic nephropathy, Proc Natl Acad Sci U S A.2019 Sep 24;116(39):19619-19625), therefore, upregulated expression of UTX and RNF114 in the kidney may contribute to the enhanced degradation of PHGDH. Furthermore, enhanced binding between UTX and RNF114 was suggested by the endogenous co-immunoprecipitation under PA-stress (Fig. 8e). Since UTX forms a complex with RNF114 that leads to PHGDH degradation, thus enhanced binding may also induce increased PHGDH degradation. However, other factors that trigger the changes in HFD conditions still await further investigation. We also discussed this issue in the revised Discussion (page 20).

8. Related to question 8, Utx and PHGDH are widely expressed in liver and adipose tissues. However, no phenotype is observed in liver and adipose tissue Utx KO mice.

What causes the tissue specificity in the kidney?

Response: We thank the reviewer for this suggestion. Recent studies showed that liver specific knockout PHGDH induced NAFLD, while adipose tissue specific knockout PHGDH did not significantly increased lipid accumulation and adipose tissue weight (Downregulation of PHGDH expression and hepatic serine level contribute to the development of fatty liver disease, *Metabolism*. 2020 102:154000; Deletion of PHGDH in adipocytes improves glucose intolerance in diet-induced obese mice, *Biochemical and Biophysical Research Communications*. 2008, 504(1) 309-314). According the Human Protein Atlas websites (<https://www.proteinatlas.org/ENSG00000124226-RNF114/tissue>) and Western blots performed in mouse tissues (shown below), the protein level of RNF114 in the kidney was obviously higher than that in the liver and the adipose tissue. which may explain the absence of obvious phenotypes of liver- or adipose-tissue specific Utx KO mice. We also described this issue in the revised Discussion (page 18).

9. In Figure 6, what causes the difference in 6c and 6f regarding the PHGDH level in the control group? In 6c, there is no change in PHGDH level with PA treatment. However, PHGDH basal level is significantly higher in the ctrl group and decreased with PA treatment in 6f. Similarly, what causes the difference in 6h and 6n regarding the fold change between ctrl and utxko group with PA treatment?

Response: We thank the reviewer for this reminder. After checking the previous Figure 6c, we realized there was an unequal sample loading between lanes as indicated by the actin. Therefore, we have re-run these gels, and the new results indicated that in the control groups, PHGDH level was decreased with PA treatment (revised Fig. 6c).

The control group in Figure 6c is a CRISPR/Cas9 control plasmid (lentiCRISPR/Cas9), whereas the control “vec” group of 6f is an overexpression plasmid (pFlag-CMV2), which may result in the differences in 6c and 6f. This information is now included in the Methods. Compared with UTX KO + CT group, the PHGDH level of UTX KO + PA group is significantly decreased. In previous 6n, compared with the Ctrl + PA group, UTX KO + PA group showed significantly decreased TG content. We apologize to mislabel the marker of statistical differences. We have performed these experiments for multiple times and obtained similar results. The representative result is provided in revised 7e.

10. In both Figure 8 and 9, where PHGDH is overexpressed or serine supplemented, how is the cellular NAD⁺/NADH NAD⁺/NADH level changed? The English grammar is weak throughout the manuscript and it needs extensive editing.

Response: In the revised manuscript, we have measured the NAD⁺/NADH level under PHGDH overexpression or serine treatment. The data suggested that PHGDH overexpression or serine administration significantly increased NAD⁺/NADH level in culture HK-2 cells (revised Supplementary Fig. 15), which is consistent with a previous report (Downregulation of PHGDH expression and hepatic serine level contribute to the development of fatty liver disease, *Metabolism*. 2020 102:154000). A native English speaker has helped to edit the language of the revised manuscript.

REVIEWERS' COMMENTS

Reviewer #1 (Remarks to the Author):

The authors have appropriately addressed all my concerns from the initial submission. The only item that remained to be addressed is the significance of the "uBUM (urinary blood urea nitrogen) added to the current version, as this reviewer is not aware of the significance of this test and the authors failed to describe the test in the methods session.

Reviewer #2 (Remarks to the Author):

Many additional experiments have been done and the manuscript has been greatly improved.

One question remained about the mechanisms for decreased glomerular hypertrophy in serine treated mice in the response to the reviewer 1. Although the authors referred the reduced lipotoxicity in tubular cells, glomerular hypertrophy in obesity-related nephropathy is associated with glomerular hyperfiltration, therefore, alteration of glomerular filtration by serine treatment may be related. How about the contribution of tubule-glomerular feedback, such as the effect of SGLT2 inhibitors to suppress glomerular hyperfiltration?

Reviewer #3 (Remarks to the Author):

The authors have addressed my concerns.

REVIEWERS' COMMENTS

Reviewer #1

The authors have appropriately addressed all my concerns from the initial submission.

The only item that remained to be addressed is the significance of the "uBUM (urinary blood urea nitrogen) added to the current version, as this reviewer is not aware of the significance of this test and the authors failed to describe the test in the methods session.

Response: The authors appreciate the reviewer for commenting “The authors have appropriately addressed all my concerns from the initial submission”. uBUN, as well as uACR, are clinical used biomarkers for obesity-related kidney disease (Biomarkers of Acute and Chronic Kidney Disease. *Annu Rev Physiol* 2019; Obesity and chronic kidney disease. *Curr Opin Pediatr* 2018; Improvement of kidney function in patients with chronic kidney disease and severe obesity after bariatric surgery: A systematic review and meta-analysis. *Nephrology (Carlton)* 2022). We have included this description in the revised Results section and provided uBUN measurement assay in the revised Methods section.

Reviewer #2

Many additional experiments have been done and the manuscript has been greatly improved. One question remained about the mechanisms for decreased glomerular hypertrophy in serine treated mice in the response to the reviewer 1. Although the authors referred the reduced lipotoxicity in tubular cells, glomerular hypertrophy in obesity-related nephropathy is associated with glomerular hyperfiltration, therefore, alteration of glomerular filtration by serine treatment may be related. How about the contribution of tubule-glomerular feedback, such as the effect of SGLT2 inhibitors to suppress glomerular hyperfiltration?

Response: We thank the reviewer for the comment that “the manuscript has been greatly improved”. We also agree with the reviewer that besides inhibiting lipotoxicity, additional mechanisms may also involve in regulating glomerular hypertrophy by serine treatment. Obesity causes ectopic lipid accumulation in the kidney, especially in tubular cells, which induces tubular hypertrophy and dysfunction. These functional and structural changes caused by obesity in the tubules could lead to glomerular hyperfiltration, which indicates communications between tubules and glomeruli (Tubuloglomerular feedback: a key player in obesity-associated kidney injury. *Am J Physiol Renal Physiol* 2022; Decreased tubuloglomerular feedback response in high-fat diet-induced obesity. *Am J Physiol Renal Physiol* 2022). In diabetes, high serum glucose level induces glomerular hyperfiltration, and also affects SGLT2 in the proximal tubules that increases glucose reabsorption; while clinical used SGLT2 inhibitors not only prevent glucose reabsorption,

but also suppress glomerular hyperfiltration during diabetes, indicating possible roles of SGLT2 inhibitors in tubule-glomerular feedback (The tubular hypothesis of nephron filtration and diabetic kidney disease. *Nat Rev Nephrol* 2020). Therefore, it is possible that serine treatment may also alter glomerular filtration *via* tubule-glomerular feedback besides its effects on lipotoxicity. We have discussed this point in the revised Discussion.